# Intracellular glycosyl hydrolase PslG shapes bacterial cell fate, signaling, and the biofilm development of *Pseudomonas aeruginosa*

Jingchao Zhang[1†], Huijun Wu[2,3†‡], Di Wang[2†], Lanxin Wang[2,3], Yifan Cui[2,3], Chenxi Zhang[1], Kun Zhao[1]*, Luyan Ma[2,3]*

[1]Frontiers Science Center for Synthetic Biology and Key Laboratory of Systems Bioengineering (Ministry of Education), School of Chemical Engineering and Technology, Tianjin University, Tianjin, China; [2]State Key Laboratory of Microbial Resources, Institute of Microbiology, Chinese Academy of Sciences, Beijing, China; [3]University of Chinese Academy of Sciences, Beijing, China

*For correspondence:
kunzhao@tju.edu.cn (KZ);
luyanma27@im.ac.cn (LM)

[†]These authors contributed equally to this work

Present address: [‡]State Key Laboratory of Petroleum Pollution Control, China National Petroleum Corporation Research Institute of Safety and Environment Technology, Beijing, China

Competing interest: The authors declare that no competing interests exist.

**Abstract** Biofilm formation is one of most important causes leading to persistent infections. Exopolysaccharides are usually a main component of biofilm matrix. Genes encoding glycosyl hydrolases are often found in gene clusters that are involved in the exopolysaccharide synthesis. It remains elusive about the functions of intracellular glycosyl hydrolase and why a polysaccharide synthesis gene cluster requires a glycosyl hydrolase-encoding gene. Here, we systematically studied the physiologically relevant role of intracellular PslG, a glycosyl hydrolase whose encoding gene is co-transcribed with 15 *psl* genes, which is responsible for the synthesis of exopolysaccharide PSL, a key biofilm matrix polysaccharide in opportunistic pathogen *Pseudomonas aeruginosa*. We showed that lack of PslG or its hydrolytic activity in this opportunistic pathogen enhances the signaling function of PSL, changes the relative level of cyclic-di-GMP within daughter cells during cell division and shapes the localization of PSL on bacterial periphery, thus results in long chains of bacterial cells, fast-forming biofilm microcolonies. Our results reveal the important roles of intracellular PslG on the cell fate and biofilm development.

## Editor's evaluation

One of the major findings related to the production of long chain polysaccharide polymers by bacteria is why there often is a hydrolytic enzyme encoded within the gene cluster encoding the biosynthetic proteins. At a basic level, these hydrolases cleave the chains to prevent toxic accumulation of intracellular polysaccharides, but it has now been discovered that a glycosyl hydrolase acting on the *Pseudomonas aeruginosa* PSL polysaccharide affects other important responses of this Gram-negative bacterium. The PslG hydrolase regulated cell length, PSL cell surface localization and signaling via cyclic-di-GMP, overall controlling accumulation of the cells into a protective biofilm.

## Introduction

Structured, surfaced-associated communities of microorganism known as biofilms are important life forms of bacteria prevailing in nature, industrial, and clinical settings (*Costerton et al., 1995*; *Stoodley et al., 2002*). In general, biofilm development involves four specific stages: attachment, microcolony formation, matured microcolonies, and dispersal. Bacteria within biofilms are embedded

in an extracellular matrix that protects bacterial cells from antibiotics, host defenses, and environmental stresses. Even though the components of biofilm matrix differ from species to species, it generally composes of exopolysaccharides, proteins, and nucleic acids (*Stoodley et al., 2002*; *Flemming and Wingender, 2010*). Exopolysaccharides are critical biofilm matrix components for many bacteria, which often promote attachment to surfaces and other cells, act as a scaffold to help maintain biofilm structure, and provide protection (*Stewart and Costerton, 2001*; *Häussler and Parsek, 2010*; *Stewart and Costerton, 2001*). Gene encoding glycosyl hydrolase is often found in gene clusters that are involved in the synthesis of exopolysaccharide (*Franklin et al., 2011*). Beyond of exopolysaccharide degradation, little is known about whether these genes affect bacterial physiology and biofilm development.

*Pseudomonas aeruginosa* is an opportunistic human pathogen that can cause life-threatening infections in cystic fibrosis (CF) patients and immune-compromised individuals (*Govan and Deretic, 1996*; *Lyczak et al., 2000*; *Ramsey and Wozniak, 2005*). *P. aeruginosa* can produce at least three different types of exopolysaccharides: alginate, PEL, and PSL (also named as Pel/Psl polysaccharide previously). Alginate is not expressed at high levels in the majority of non-CF isolates (*Colvin et al., 2012*), whereas PSL is expressed by most *P. aeruginosa* natural and clinical isolates (*Colvin et al., 2012*; *Ma et al., 2012*; *Jennings et al., 2021*). PEL is significant for biofilm formation when PSL cannot be synthesized (*Jennings et al., 2015*). In *P. aeruginosa* PAO1, PSL is a primary scaffold matrix component that can form a fibre-like matrix to enmesh bacteria within a biofilm (*Colvin et al., 2012*; *Ma et al., 2009*; *Wang et al., 2013*). PSL has shown multiple functions in the biofilm formation of PAO1. For example, PSL can act as a 'molecular glue' to promote bacteria cell-cell and cell-surface interactions (*Ma et al., 2006*; *Ma et al., 2009*). PSL trails on a surface guide bacteria exploration and microcolony formation (*Zhao et al., 2013*). Moreover, PSL can also work as a barrier to protect bacteria from antibiotics and phagocytic cells (*Mishra et al., 2012*; *Billings et al., 2013*; *Tseng et al., 2013*). Interestingly, PSL can function as a signal to stimulate biofilm formation through affecting intracellular signal molecule cyclic-di-GMP (c-di-GMP) (*Irie et al., 2012*).

PSL is synthesized by *psl* operon, containing 15 co-transcribed genes (*pslABCDEFGHIJKLMNO*) (*Byrd et al., 2009*). PslG is a glycosyl hydrolase that encoded by *pslG* wthin *psl* operon. PslG has been shown to degrade PSL in vitro or within biofilm matrix which is released from dead bacteria (*Yu et al., 2015*; *Zhao et al., 2018*), hence it can inhibit biofilm formation and disrupt a formed biofilm at a nanomolar concentration (*Yu et al., 2015*; *Baker et al., 2016*). PslG was first thought to be essential for PSL synthesis, since deletion of *pslG* gene led to a loss of PSL production (*Byrd et al., 2009*). Later studies found that deletion of *pslG* in the earlier work has a polar effect on the expression of *pslH* and thus resulted in the loss of PSL production, and absence of *pslG* itself did not result in a complete loss of PSL production per se, but led to a less production of PSL and reduced bacterial initial attachment compared with PAO1 (*Baker et al., 2015*; *Wu et al., 2019*). PslG is localized mainly at the inner membrane and some in the periplasm. PslA, PslD, and PslE help PslG anchoring in the inner membrane, which is critical for PslG to be involved in the biosynthesis of PSL (*Wu et al., 2019*). In addition, the glycoside hydrolytic activity of PslG is also important for PSL production and the key amino acid residues for this activity are E165 and E276 (*Wu et al., 2019*; *Yu et al., 2015*).

C-di-GMP is an important second messenger controlling a wide range of cellular processes in many bacteria, such as motility, cell differentiation, biofilm formation and production of virulence factors (*Römling et al., 2013*). Reports have shown that c-di-GMP is asymmetrically distributed among daughter cells upon bacterial cell division and the asymmetric division on surfaces produces specialized cell types, a spreader for dissemination and a striker for local tissue damage (*Christen et al., 2010*; *Laventie et al., 2019*). It has not been investigated whether an intracellular glycoside hydrolase would affect the c-di-GMP level.

In this work, aiming to study the effect of PslG on the cell fate and biofilm development of *P. aeruginosa* at the single cell level, we systematically studied the *pslG* in-frame deletion mutants by employing a high-throughput bacterial tracking technique (*Zhao et al., 2013*). The morphology and motility behavior of bacterial cells in the course of biofilm development were analyzed at the single-cell level. Using pCdrA::*gfp* reporter, the c-di-GMP level of each cell was also monitored. Microscopically, the attachment behavior of cells on the microtiter surfaces and the pellicles formed at the air-liquid interface were also characterized. Our data suggest that lacking of *pslG* impacts cell morphology, the signaling function of PSL, the c-di-GMP distribution and bacteria distribution within

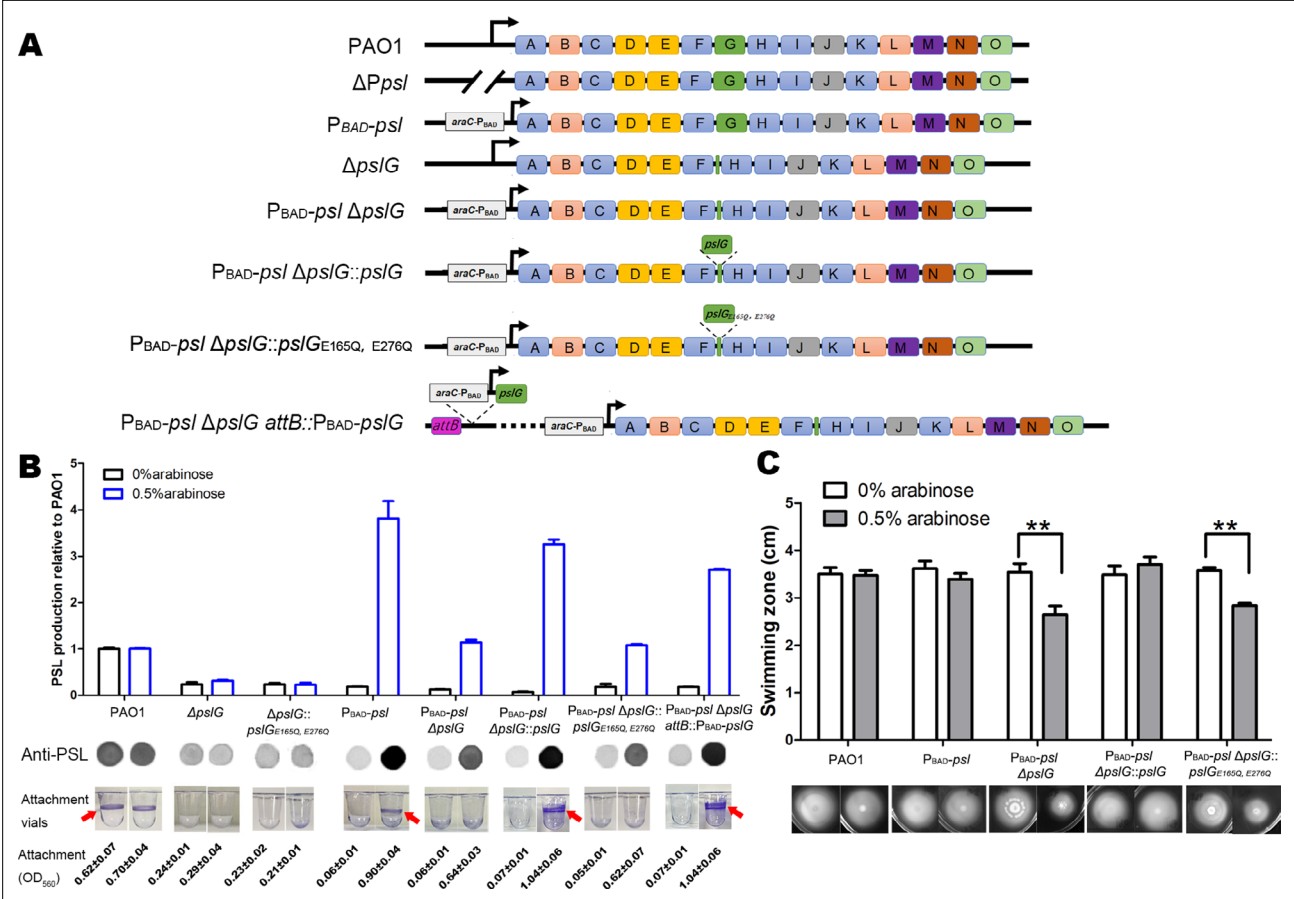

**Figure 1.** Inducing PSL production in Δ*pslG* background cannot recover its defects on bacterial initial attachment and yet affects swimming motility. (A): A schematic of the *psl* operon in PAO1 and its corresponding mutants used in this study. Genes *pslA-O* are shown in boxes (not to scale). Angled lines represent the extent of deleted sequence, and black arrows indicate transcriptional start sites (not to scale). (B): The Psl production of tested strains inducing with 0% or 0.5% arabinose. The amount of Psl was determined by immune-dot blotting and normalized to the level of PAO1. A representative of dot blotting as well as corresponding microtiter dish wells and their crystal violet (CV) reading (OD$_{560}$) posted CV staining in attachment assay were shown under each corresponding column. Arrows indicate the ring at air-liquid interface. (C): The swimming motility of tested strains inducing with 0% or 0.5% arabinose. The corresponding image of swimming zone was shown under each column. Statistical significances were measured using student's t-test (**, p < 0. 01).

The online version of this article includes the following source data and figure supplement(s) for figure 1:

**Source data 1.** *Figure 1B, C* source data.

**Figure supplement 1.** The influence of Δ*pslG* on T4P-driven twitching motility.

a biofilm. Based on our results together with those in literature, a model is proposed to understand the role of *pslG* in the biofilm development.

## Results

### Δ*pslG* strains cannot form rings on microtiter dish wells even when PSL production is induced to the wild-type level

Our previous study showed that the *pslG* in-frame deletion mutant (was named as Δ*pslG*2 by *Wu et al., 2019* hereafter termed as Δ*pslG*) decreased the production of PSL and bacterial initial attachment on the microtiter surface (*Wu et al., 2019*). To know whether the attachment defect of Δ*pslG* mutant is due to PSL reduction, we replaced the promoter of *psl* operon by P$_{BAD}$ promoter in Δ*pslG* background, resulting in a PAO1-derived Psl-inducible Δ*pslG* strain (named as P$_{BAD}$-*psl*Δ*pslG*), whose PSL production can be induced by the concentration of arabinose. When induced with 0.5% arabinose, P$_{BAD}$-*psl*Δ*pslG* strain produced similar amount of PSL as that of PAO1 (*Figure 1B*). However, it

was not able to form a ring at the air-liquid interface as that seen in either PAO1 or P$_{BAD}$-*psl* strain (referred as WFPA801 previously, *Ma et al., 2006*) under 0.5% arabinose induction (*Figure 1B*, rings are indicated by an arrow). The ring formation can be recovered by either P$_{BAD}$-*pslG* inserted in chromosome *attB* site (P$_{BAD}$-*psl* Δ*pslG attB::pslG* in *Figure 1B*) or *pslG* knocked into the Δ*pslG* mutants at the original location of *pslG* (P$_{BAD}$-*pslΔpslG::pslG* in *Figure 1B*). Whereas PslG$_{E165Q, E276Q}$ (two key hydrolytic active sites of PslG were mutated) cannot complement this phenotype (*Figure 1B*). These results indicate that PslG and its glycoside hydrolytic activity are important for the ring formation at the air-liquid interface.

Flagella and Type IV pili (T4P) are also important for the initial attachment of *P. aeruginosa* (*O'Toole and Kolter, 1998*; *Bruzaud et al., 2015*). We then tested the flagellum-driven swimming motility and T4P-mediated twitching motility of P$_{BAD}$-*pslΔpslG* strain to evaluate the function of flagella and T4P. Without arabinose induction, P$_{BAD}$-*pslΔpslG* strain exhibited similar swimming ability and twitching motility as did PAO1 and P$_{BAD}$-*psl* (*Figure 1C* and *Figure 1—figure supplement 1*). However, with 0.5% arabinose induction, P$_{BAD}$-*pslΔpslG* strain (having a wild type level of PSL production, *Figure 1B*) showed reduced swimming zone compared to that of either PAO1 or P$_{BAD}$-*psl* (*Figure 1C*). T4P-mediated twitching motility of P$_{BAD}$-*pslΔpslG* was not affected under conditions with or without arabinose (*Figure 1—figure supplement 1C*). These results demonstrate that *pslG* deletion does not affect the function of flagella or T4P directly, yet inducing PSL production in P$_{BAD}$-*pslΔpslG* attenuates the swimming motility with no effect on bacterial growth (*Figure 1—figure supplement 1D*), which might impact its attachment phenotype. Taken together, our results show that increasing PSL production in P$_{BAD}$-*pslΔpslG* could not rescue its defect on attachment, suggesting that Δ*pslG* might have multiple effects on bacterial physiology.

## Δ*pslG* impacts the bacterial distribution and maximum thickness of pellicles

We then investigated the effect of Δ*pslG* on the biofilms formed at the air-liquid interface, termed as pellicles, by using confocal laser scanning microscopy. The total pellicle biomass of Δ*pslG* is similar to that of PAO1 after 24 hr growth, although Δ*pslG* produced much less PSL and had defect on initial attachment (*Figures 1, 2A and C*). However, Δ*pslG* has significant higher maximum thickness than that of PAO1 (*Figure 2B*). In addition, there are less bacteria in each section image of Δ*pslG* pellicles compared to that of PAO1 (*Figure 2D*, left and middle panel). The PSL matrix in Δ*pslG* pellicles shows weaker fluorescent intensities than that of PAO1 (*Figure 2D*, middle and right panels), which is consistent with their corresponding PSL production. In spite of that, the fibre-like PSL can be detected in the pellicles of Δ*pslG*, which have a radial pattern as previously described for PAO1 pellicles (*Figure 2D*, middle panels) (*Wang et al., 2013*). These results suggest that the *pslG* deletion might impact bacterial distribution within biofilms.

## Single-cell tracking analysis indicates that loss of PslG or its glycoside hydrolytic activity promotes microcolony formation in flow-cell systems

To further understand the effect of Δ*pslG*, by employing bacterial tracking techniques, we observed Δ*pslG* cell behavior at the single-cell level. *Figure 3A* shows the surface coverage obtained by all tracked bacterial trajectories for a specific time period during microcolony formation. Red color indicates the surface area that has been visited by bacteria, while black color indicates a 'fresh' surface area that has never been visited. Bacterial cells are shown in blue. Under the same total bacterial visits (marked as *N* in *Figure 3A*), the difference in the surface coverage between Δ*pslG* and PAO1 is not obvious at *N*~10,000. As *N* increases, the surface coverage of Δ*pslG* is clearly less than that of PAO1 (*Figure 3A*). At *N*~100,000, Δ*pslG* has a surface coverage of 52% ± 6% while PAO1 has 81% ± 10%. Compared with PAO1, the less efficiency in covering the surface leads to a more non-uniform bacterial visit distribution for Δ*pslG* (*Figure 3B*). Correspondingly, by fitting the distribution of bacterial visits with a power law, different power law exponents were obtained (see one example in *Figure 3C*). The averaged power law exponents of three repeats are –2.9 ± 0.1 for PAO1 and –2.5 ± 0.1 for Δ*pslG*. Such differences in bacterial visits distribution resulted that the time required to observe a visible microcolony (defined as clusters of more than 30 cells in this study, marked by dash lines in *Figure 3D*) in the field of view is shorter for Δ*pslG* (5.9 ± 1.7 hr) than that for PAO1 (8.7 ± 2.4 hr). In addition, after 10 hr of growth, compared with PAO1, Δ*pslG* formed more microcolonies in the field of view

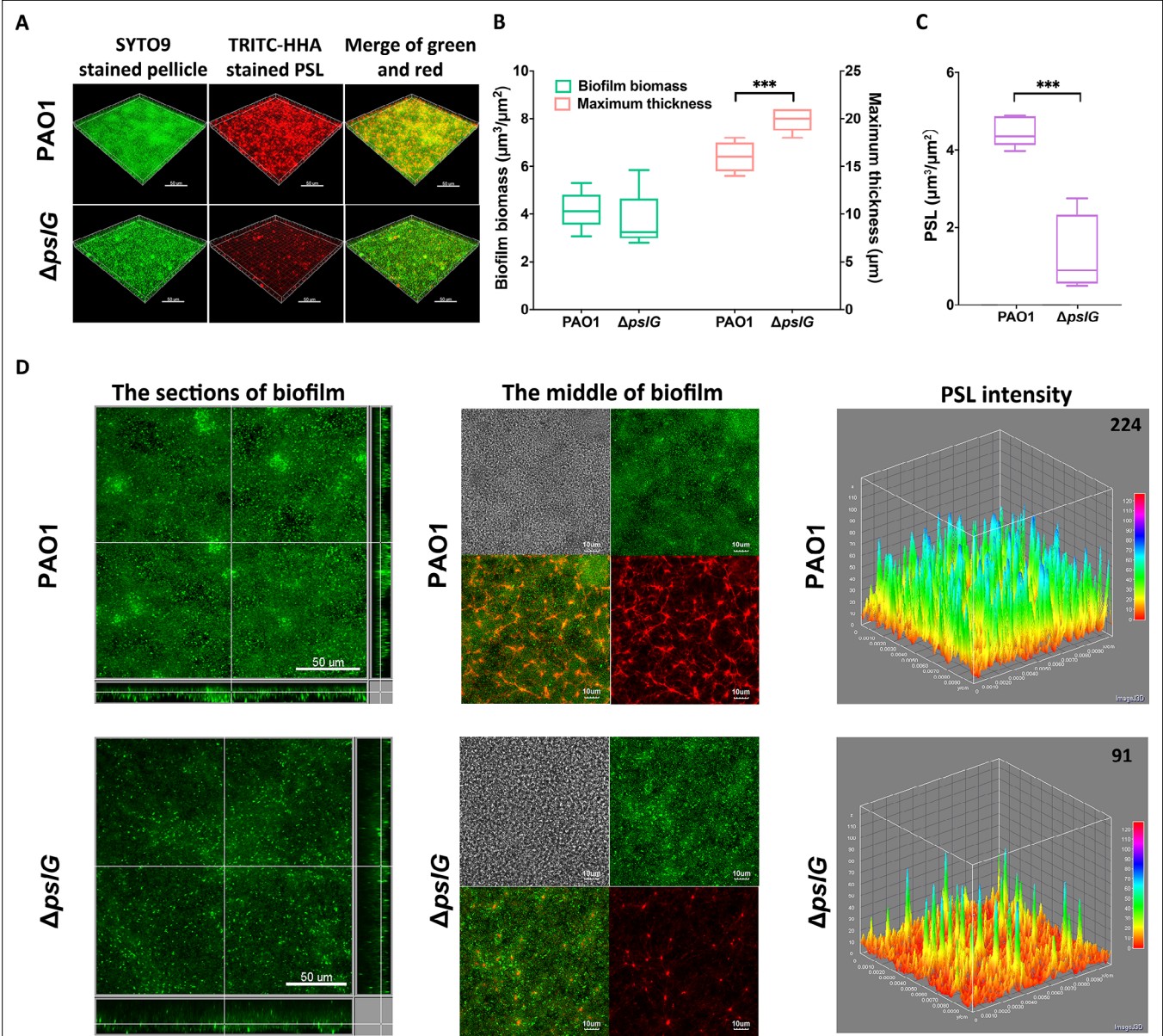

**Figure 2.** Comparison of pellicles formed by PAO1 and Δ*pslG* mutant. (**A**): Three-dimensional images of 24 hr air-liquid interface biofilms (pellicles) formed by PAO1 and Δ*pslG*. (**B**): Biofilm biomass and maximum thickness of PAO1 or Δ*pslG* strain. (**C**): PSL in pellicles of PAO1 and Δ*pslG*. (**D**): Typical section images of pellicles formed by PAO1 and Δ*pslG*. Left panel, section images showed the top-down view (square) and side view (rectangle) of corresponding pellicles. Middle panel, section images at the middle of corresponding pellicles. The distribution of bacteria (green), the fibre-like PSL matrix (red) and corresponding DIC images (grey) were shown. Right panel, PSL fluorescence intensity in corresponding biofilm images shown in the middle panel (the average intensity of PSL in per µm³ biofilm is shown in the upper right corner). Green, SYTO9 stained bacteria, Red, TRITC-HHA stained PSL. Statistical significances were measured using student's t-test (***, p < 0.001 when compared to PAO1). Scale bar: 50 µm for A and the left panel in (**D**); 10 µm for the middle panel in D.

The online version of this article includes the following source data for figure 2:

**Source data 1.** *Figure 2B, C* source data.

(*Figure 3E*). The microcolony formation phenotype of Δ*pslG* mutants can be reverted back to WT-like when they are complemented with PslG by knocking *pslG* into the Δ*pslG* mutants at the original location of *pslG*, namely Δ*pslG::pslG*. As illustrated in *Figure 3*, at *N*~100,000, the surface coverage of Δ*pslG::pslG* reaches to 70% ± 7% and the power law exponent of the bacterial visit distribution is –3.2 ± 0.13 (averaged over three repeats), both are closer to those of PAO1 than those of Δ*pslG*.

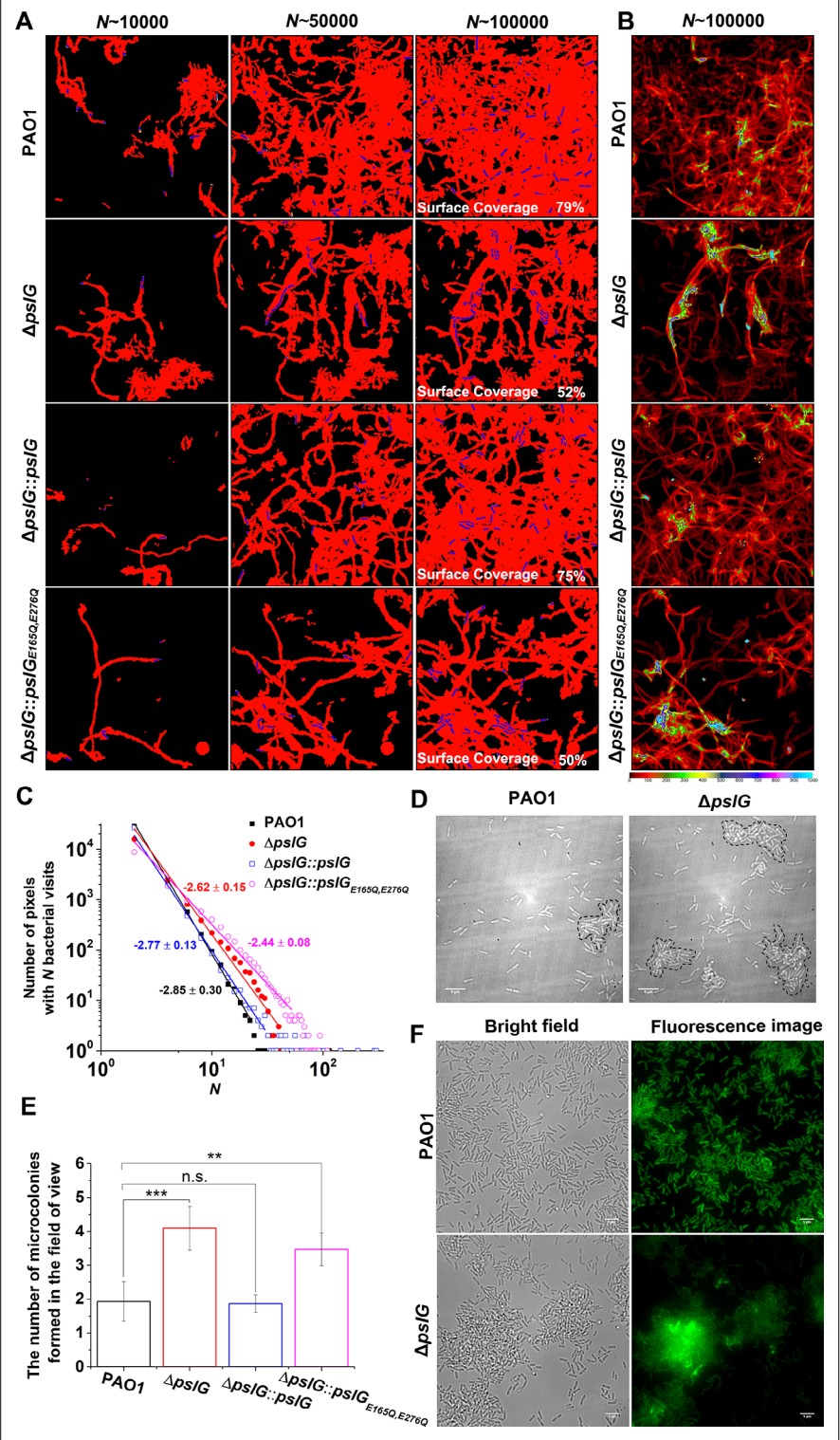

**Figure 3.** Effects of *pslG* on the formation of microcolonies in flow-cell systems. (A): Surface coverage maps at a total of 10,000, 50,000, and 100,000 bacterial visits for PAO1, Δ*pslG*, Δ*pslG::pslG*, and Δ*pslG::pslG*$_{E165Q,E276Q}$ cells. Red color indicates the surface area that has been visited or contaminated, while black color indicates a 'fresh' surface area. Bacteria in the current frame are shown in blue. The surface coverage in the figure is the result of an experiment. (B): The intensity map of bacterial visits at $N \sim 100,000$. The color scale of black to cyan represents bacterial visits of 0–1,000. (C): The graph displays one measurement result for the visit frequency distributions of PAO1, Δ*pslG*, Δ*pslG::pslG* and Δ*pslG::pslG*$_{E165Q,E276Q}$ at $N \sim 100,000$. The slope in the figure is the fitting result

*Figure 3 continued on next page*

*Figure 3 continued*

of an experiment. (D): Examples of microcolonies (enclosed by dash lines) formed by PAO1 and Δ*pslG* cells, respectively, cultured in a flow cell for about 8 hr. (E): The number of microcolonies in the field of view formed by PAO1, Δ*pslG*, Δ*pslG::pslG* and Δ*pslG::pslG*$_{E165Q,E276Q}$ at 10 hr after inoculation in a flow cell. The number (**N**) of frames analyzed are 14, 43, 52, and 51 for PAO1, Δ*pslG*, Δ*pslG::pslG*, and Δ*pslG::pslG*$_{E165Q,E276Q}$, respectively. Error bars represent standard deviations of the means. Statistical significances were measured using one-way ANOVE. n.s., not significant; *$p < 0.05$; **$p < 0.001$; ***$p < 0.0001$. F. Snapshots taken at 10 hr after inoculation in a flow cell, showing the microcolonies formed by Δ*pslG* and PAO1. Bacteria were tagged by GFP. Fluorescence images and corresponding bright-field images were shown. Scale bar, 5 μm.

The online version of this article includes the following source data and figure supplement(s) for figure 3:

**Source data 1.** *Figure 3C* source data.

**Source data 2.** *Figure 3E* source data.

**Figure supplement 1.** The microcolonies formed in flow-cell systems by tested strains.

**Figure supplement 2.** Effects of *pslG* deletion in P$_{BAD}$-*psl* background on the formation of microcolonies in flow-cell systems.

**Figure supplement 3.** The microcolony formation and initial attachment of ΔP*pel*, and Δ*pslG*ΔP*pel* strains.

The number of microcolonies formed by Δ*pslG::pslG* in the field of view after 10 hr of growth is also similar to that of PAO1 and less than that of Δ*pslG*. The fast microcolony formation phenotype of Δ*pslG* mutant is also able to be complemented by PslG that is expressed from plasmid (***Figure 3—figure supplement 1***). However, Δ*pslG::pslG*$_{E165Q,E276Q}$, in which the key glycoside hydrolytic activity sites are mutated cannot recover WT-like phenotype (***Figure 3*** and ***Figure 3—figure supplement 1***), suggesting the importance of PslG hydrolytic activities. Complementation tests were also performed in strains from P$_{BAD}$-*psl* background, and similar trends were observed (***Figure 3—figure supplement 2***). PEL has been shown to play a role in bacterial aggregation (***Jennings et al., 2021***), thus we also tested PEL-negative strains in wild type (ΔP$_{pel}$) or the Δ*pslG* background (Δ*pslG*ΔP$_{pel}$). The phenotype of Δ*pslG*ΔP$_{pel}$ was similar to that of Δ*pslG*, suggesting little contribution of PEL on the fast microcolony formation of Δ*pslG* (***Figure 3*** and ***Figure 3—figure supplement 3***). Tracking GFP-tagged bacteria in a flow cell also shows that the PAO1 biofilms tend to spread on surfaces (***Figure 3F*** and ***Video 1***) while Δ*pslG* cells tend to accumulate and form microcolonies with strong fluorescence intensity (***Figure 3F*** and ***Video 2***). This is consistent with the results of bacterial visit distribution map shown in ***Figure 3B***. Taken together, these results indicate that loss of *pslG* promotes microcolonies formation by changing

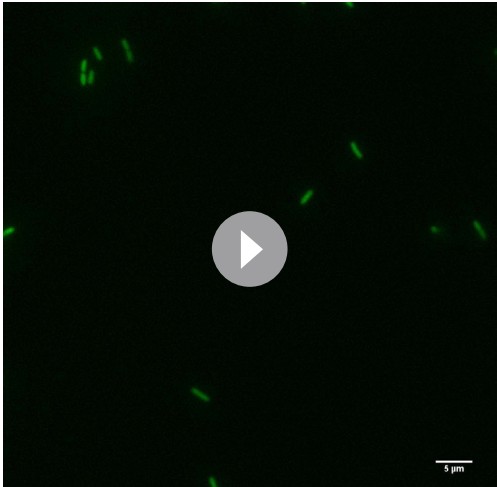

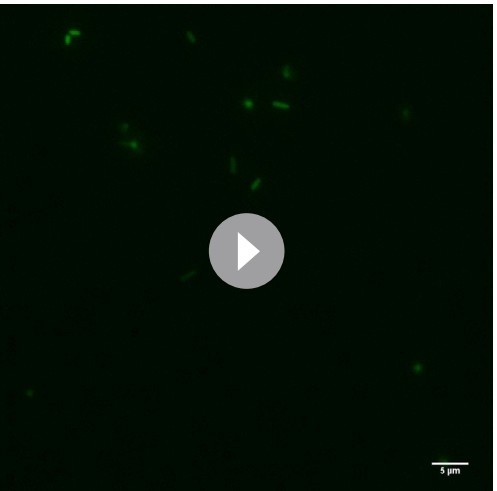

**Video 1.** An example video of tracking the biofilm formation of Gfp-tagged PAO1 cells. The video was taken at a frame interval of 10 min for 10 hr and was played back at 5 fps. Scale bar, 5 μm.

https://elifesciences.org/articles/72778/figures#video1

**Video 2.** An example video of tracking the biofilm formation of Gfp-tagged Δ*pslG* cells. The video was taken at a frame interval of 10 min for 10 hr and was played back at 5 fps. Scale bar, 5 μm.

https://elifesciences.org/articles/72778/figures#video2

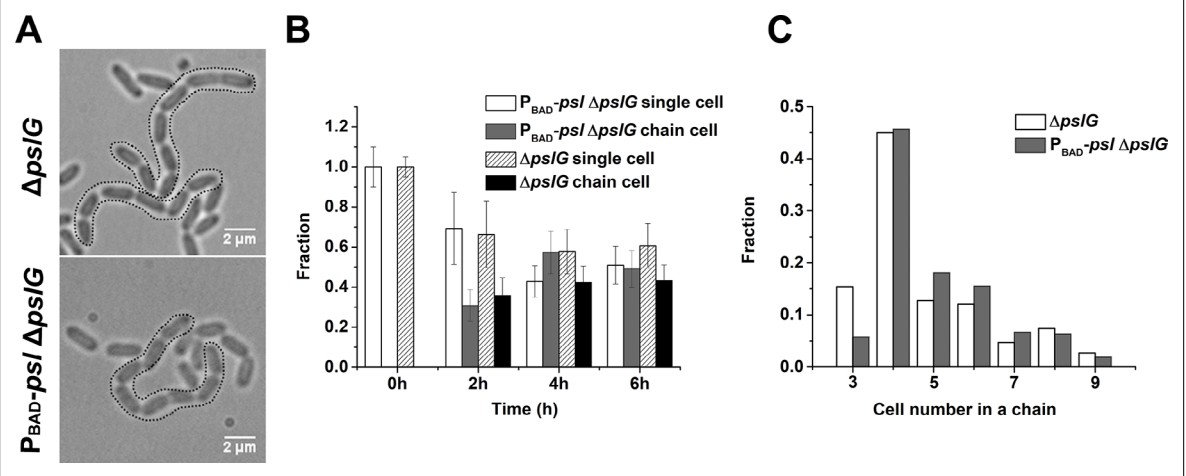

**Figure 4.** Characterization of long bacterial chains of Δ*pslG* strains. (A): Examples of long bacterial chains (indicated by black dotted outlines) formed by Δ*pslG* and P$_{BAD}$-*psl*Δ*pslG* cells. (B): The faction of single isolated bacterial cell and cell in chains at different time points after inoculation in a flow cell. The number of analyzed picture in each strain is n = 88 (about 3200 cells) for P$_{BAD}$-*psl*Δ*pslG* and n = 87 (about 3500 cells) for Δ*pslG*. (C): The number distribution of cells consisted in a chain. The number of analyzed cells is n = 301 for P$_{BAD}$-*psl*Δ*pslG* and n = 322 for Δ*pslG*. Scale bar, 2 μm.

The online version of this article includes the following source data and figure supplement(s) for figure 4:

**Source data 1.** *Figure 4B* source data.

**Source data 2.** *Figure 4C* source data.

**Figure supplement 1.** Examples of PAO1, Δ*pslG*, P$_{BAD}$-*psl* and P$_{BAD}$-*psl*Δ*pslG* cells grown under 0% arabinose and 0.5% arabinose.

the surface exploration of bacteria during microcolony formation, a phenomenon that has been reported mostly for strains producing a high level of PSL (*Zhao et al., 2013*).

## T4P-driven motility is not the main factor in promoting the microcolony formation of Δ*pslG* strains in flow-cell systems

T4P-driven motilies on surface, such as walking and crawling, affect microcolonies formation in flow-cell systems (*Conrad et al., 2011*). To investigate why Δ*pslG* promotes microcolonies formation, we calculated the twitching speed of bacterial cells. To minimize the effect due to possible different production of PSL, we compared the measurements between PAO1 and P$_{BAD}$-*psl*Δ*pslG* under 0.5% arabinose, under which both strains show relatively similar production of PSL (*Figure 1*). The results show a slightly reduced average speed and a higher crawling percentage of P$_{BAD}$-*psl*Δ*pslG* cells compared with PAO1 (*Figure 1—figure supplement 1A*, B). But such differences are not statistically significant (*P* = 0.29), indicating that the twitching motility may not be the main factor in promoting the microcolony formation.

## Δ*pslG* shapes the localization of PSL on bacterial periphery, leading to long chains of bacterial cells that are connected by PSL

During bacterial tracking in flow cells, we frequently observed long chains of bacterial cells in the Δ*pslG* strain or P$_{BAD}$-*psl*Δ*pslG* strain when PSL production was induced with arabinose

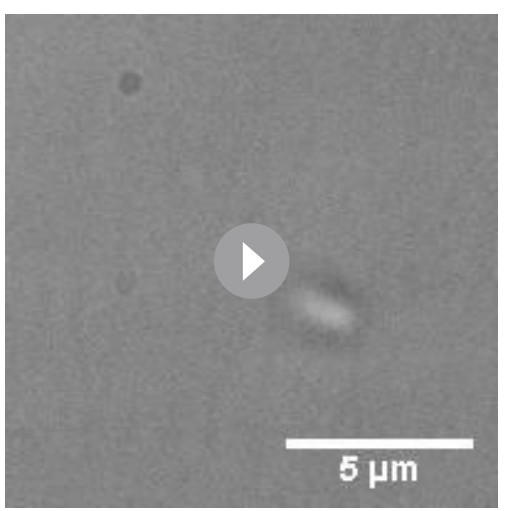

**Video 3.** An example video of the formation of P$_{BAD}$-*psl*Δ*pslG* long cell-chain. The video was taken at a frame interval of 5 min for 3.5 hr and was played back at 5 fps. Scale bar, 5 μm.
https://elifesciences.org/articles/72778/figures#video3

(*Figure 4A*), which typically started to appear 1~2 hr after inoculation of bacteria into a flow cell under tested conditions and could reach to about 50% of cell population at a later time (*Figure 4B*). Such long chains of cells were not observed in strains that have intact *pslG* or when PSL production was not induced in P$_{BAD}$-*pslΔpslG* (*Figure 4—figure supplement 1*). In a typical cell division process, two daughter cells will be disconnected from each other when the formation of septum is completed. However, in *pslG* deletion mutants, the two daughter cells could not separate into physically disconnected progenies, leading to a cell chain (See one example of P$_{BAD}$-*pslΔpslG* in *Video 3*). The length of chains varied. Among all the observed chains, the chains consisting of 4 cells were observed most frequently, which is true both in Δ*pslG* and in P$_{BAD}$-*pslΔpslG* strains (*Figure 4C*). These bacterial cell chains can grow as cells continue to divide, yet some can also be broken by bending of chains (*Video 3*), suggesting that the bacterial chains are not connected by septum. In addition, bacterial cell chains were also observed in liquid culture of Δ*pslG* mutants (data not shown), indicating that bacterial adhering on surfaces is not required for the formation of bacterial cell chains.

To investigate whether PSL has any contribution on the formation of such long bacterial chains, we stained PSL in the biofilm formed in flow cells by FITC-HHA (green fluorescence dyes FITC labeled lectin HHA). The PSL of Δ*pslG* strains is tightly associated with bacteria compared to PAO1 and P$_{BAD}$-*psl* with arabinose (*Figure 5A* and *Figure 5—figure supplement 1*). We also used cell membrane stain FM4-64 to help locate cell periphery and septum. Strikingly, strong PSL signal is often found around septa in Δ*pslG* strains and strains with catalytic site mutation (P$_{BAD}$-*pslΔpslG::pslG*$_{E165Q, E276Q}$ or Δ*pslG::pslG*$_{E165Q, E276Q}$), which barely observed in strains with wild type *pslG* (such as PAO1 and P$_{BAD}$-*psl*) under the same growth conditions (*Figure 5A* and *Figure 5—figure supplement 1*).

From the PSL staining results, we speculate that PSL might help to connect bacterial cells together to form long bacterial chains. To test this hypothesis, we first tested whether the P$_{BAD}$-*pslΔpslG* could form long bacterial chains when PSL is not produced. Under a culture condition without arabinose, the transcription of *psl* operon in P$_{BAD}$-*pslΔpslG* is not induced, no bacterial chains were observed as shown in *Figure 4—figure supplement 1*, suggesting that PSL production is required for the formation of long bacterial chains. Next, we treated the long bacterial chains with purified PslG, which has been shown to be able to disaggregate microcolonies and matured biofilms by hydrolyzing PSL (*Yu et al., 2015*). At 4 min after addition of exogenous PslG, the long chains of bacterial cells were seen clearly to start to be broken up, and they were completely disconnected into single cells after 12 min of PslG treatment (*Figure 5B*) (*Videos 4 and 5*). Bacterial chains can be separated by exogenous PslG within a few minutes further confirmed that the bacterial chains were not connected by septum. These results together with the fact that PSL is a 'sticky' exopolysaccharide suggest that bacterial cells are frequently connected by PSL in Δ*pslG* mutant strains, leading to the long bacterial cell chains.

This long-cell-chain phenotype can be complemented by wild type PslG, but not hydrolytic activity sites mutation PslG$_{E165Q, E276Q}$ (*Figure 5—figure supplement 2*), suggesting that the hydrolytic activity of PslG plays a key role on the formation of long bacterial cell chains. To check whether exopolysaccharide PEL and alginate play roles on the observed phenotype induced by *pslG* deletion, we also tested PAO1-derived alginate-negative strain Δ*algD* as well as PEL-negative strains ΔP$_{pel}$ and Δ*pslG*ΔP$_{pel}$. The phenotype of either ΔP$_{pel}$ or Δ*algD* is similar to that of PAO1, whereas Δ*pslG*ΔP$_{pel}$ exhibits a similar phenotype as Δ*pslG* strains, for bacterial chains formation, microcolonies formation, and initial attachment (*Figure 5—figure supplement 3*, *Figure 3—figure supplement 3*, *Figure 1—figure supplement 1C*). These results indicate that PEL and alginate may have little effect on the observed phenotype induced by *pslG* deletion in this study. This data also imply that the long bacterial chains might be a contributor for Δ*pslG* strains to promote microcolony formation.

## Lack of PslG or its hydrolytic activity has effects on the fate and C-di-GMP distribution of daughter cells during cell division

C-di-GMP is a critical intracellular signal molecule that affects a variety of cell activities including cell motility, cell fate after division and biofilm formation. We then monitored the c-di-GMP level of cells for each cell division event by employing pCdrA::*gfp* as a reporter (*Irie et al., 2012*). Cells with high c-di-GMP levels show strong fluorescence intensity as previously described (*Irie et al., 2012*). We focused on the first bacterial division events after cells attached on the surface and analyzed the fluorescence intensity (corresponding to c-di-GMP levels) in two daughter cells right after division (*Figure 6A*). By comparing the fluorescence intensity of each daughter cell relative to its mother

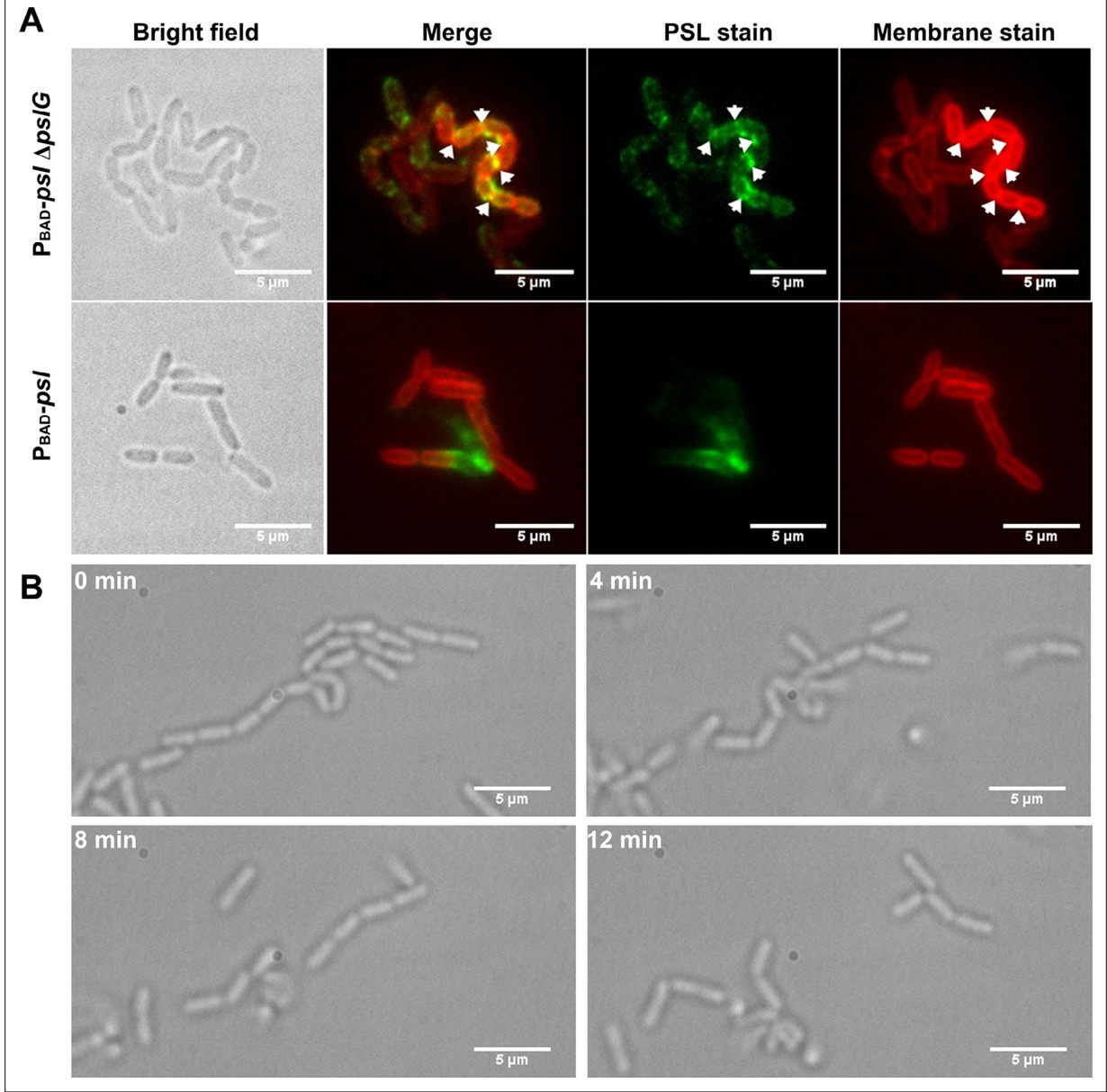

**Figure 5.** Cells in bacterial chains are connected by PSL and can be disassembled by PslG supplied exogenously. (A): Fluorescence staining of long bacterial chains formed by P$_{BAD}$-psl$\Delta$pslG cells. Staining of control samples of P$_{BAD}$-psl is also shown. Green shows PSL stained by FITC-HHA, and red shows the bacterial cell membranes stained by FM4-64. Arrows in the PSL stain image indicate the bright spotted PSL locations, and arrows in the membrane stain image show the septum locations. (B): Time–lapse images show the break-up process of long bacterial chains of P$_{BAD}$-psl$\Delta$pslG when PslG was supplied. Scale bar, 5 µm.

The online version of this article includes the following figure supplement(s) for figure 5:

**Figure supplement 1.** Representative PSL and membrane double staining images of PAO1 and pslG mutants grown in flow-cell systems.

**Figure supplement 2.** Bacterial morphology of PAO1/pHERD20T, $\Delta$pslG/pHERD20T, $\Delta$pslG/pG, $\Delta$pslG/pGDM, $\Delta$pslG::pslG, $\Delta$pslG::pslG$_{E165Q,E276Q}$, P$_{BAD}$-psl$\Delta$pslG::pslG and P$_{BAD}$-psl$\Delta$pslG::pslG$_{E165Q,E276Q}$ cells grown 0.1% arabinose.

**Figure supplement 3.** The formation of long bacterial chains by $\Delta$Ppel, $\Delta$pslG$\Delta$Ppel, and $\Delta$algD cells.

cell, the division events can be classified into three types: none of the daughter cells becomes bright (none-bright), one daughter cell becomes bright (one-bright) and both of daughter cells become bright (two-bright). The results show that none-bright type is observed most frequently in all tested strains, which has an occurrence probability of ~ 60% for PAO1, ~ 58% for $\Delta$pslG, and ~ 52% for P$_{BAD}$-psl$\Delta$pslG. Interestingly, both $\Delta$pslG mutants show a relatively higher probability of two-bright type

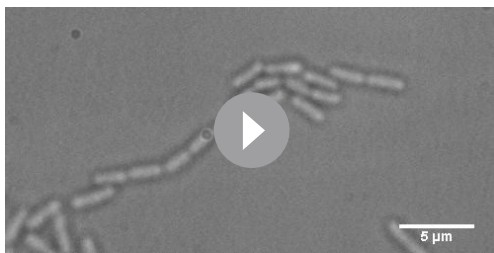

**Video 4.** An example video of the degradation of P_BAD-*pslΔpslG* long cell-chain. The video was taken at a frame interval of 1 min for 0.5 hr and was played back at 5 fps. Scale bar, 5 μm.

https://elifesciences.org/articles/72778/figures#video4

(~17% for P_BAD-*pslΔpslG* strain, ~ 16% for *ΔpslG* strain) than that of PAO1 (~11%) (*Figure 6B* and *Figure 6—figure supplement 1*). In addition, the catalytic site mutated strains show a phenotype similar to *ΔpslG* strains, have a relatively higher probability of two-bright type (~19% for both *ΔpslG::pslG*_E165Q, E276Q and P_BAD-*pslΔpslG::pslG*_E165Q, E276Q strain) (*Figure 6B*). Thus, compared with PAO1, *ΔpslG* cells and bacterial cells with the catalytic site mutated PslG would have a higher probability to have both daughter cells with high c-di-GMP levels. During asymmetric divisions, daughter cells with high c-di-GMP levels keep staying on the surface and daughter cells with low c-di-GMP levels tend to move away (*Christen et al., 2010*; *Laventie et al., 2019*). Therefore, both daughter cells with high c-di-GMP levels might enhance bacterial stay on the surface and alter their movement pattern. In addition,

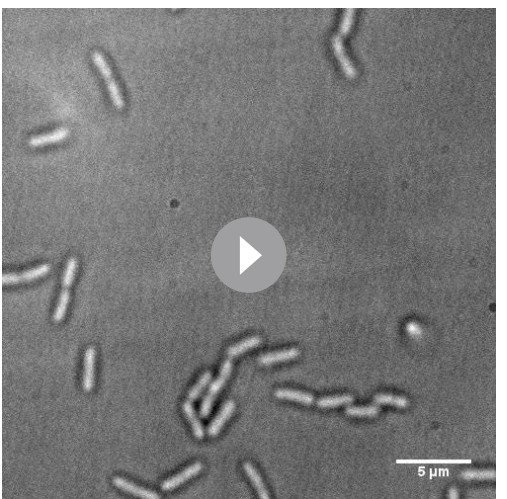

**Video 5.** An example video of the degradation of *ΔpslG* long cell-chain. The video was taken at a frame interval of 1 s for 10 min and was played back at 5 fps. Scale bar, 5 μm.

https://elifesciences.org/articles/72778/figures#video5

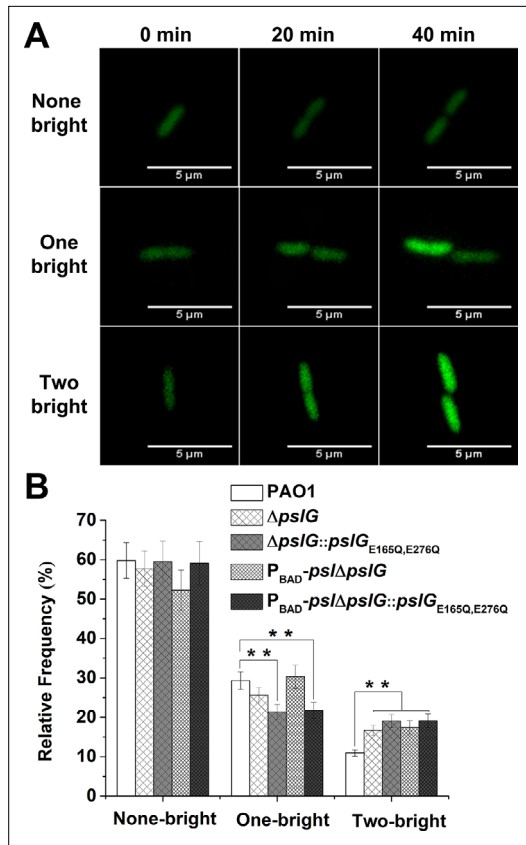

**Figure 6.** Lacking of PslG or its hydrolytic activity has effects on the fate and c-di-GMP distribution of daughter cells. (A): Three types of cell division based on fluorescence intensity changes of daughter cells relative to that of their mother cell: none of daughter cells becomes bright (none-bright), one daughter cell becomes bright (one-bright), and both daughter cells become bright (two-bright). Examples given are PAO1 cells. The fluorescence is from pCdrA::*gfp*, which acts as a reporter for the c-di-GMP level of cells. (B): The measured probability of three types of division in PAO1, *ΔpslG*, P_BAD-*pslΔpslG*, *ΔpslG::pslG*_E165Q,E276Q, and PBAD-*pslΔpslG::pslG*_E165Q,E276Q. The total number of analyzed division events from more than three repeats is n = 174 for PAO1, n = 168 for *ΔpslG*, n = 109 for P_BAD-*pslΔpslG*, n = 131 for *ΔpslG::pslG*_E165Q,E276Q, and n = 115 for PBAD-*pslΔpslG::pslG*_E165Q,E276Q. Statistical significances were measured using one-way ANOVA. n.s., not significant; **p < 0. 01. Scale bar, 5 μm.

The online version of this article includes the following source data and figure supplement(s) for figure 6:

**Source data 1.** *Figure 6B* source data.

**Figure supplement 1.** The ratio of fluorescence intensity ( = Idau/Imot) of each daughter cell for each division event.

a high level of c-di-GMP enhances PSL production. An earlier work has shown that cells form a microcolony through a PSL-based rich-get-richer

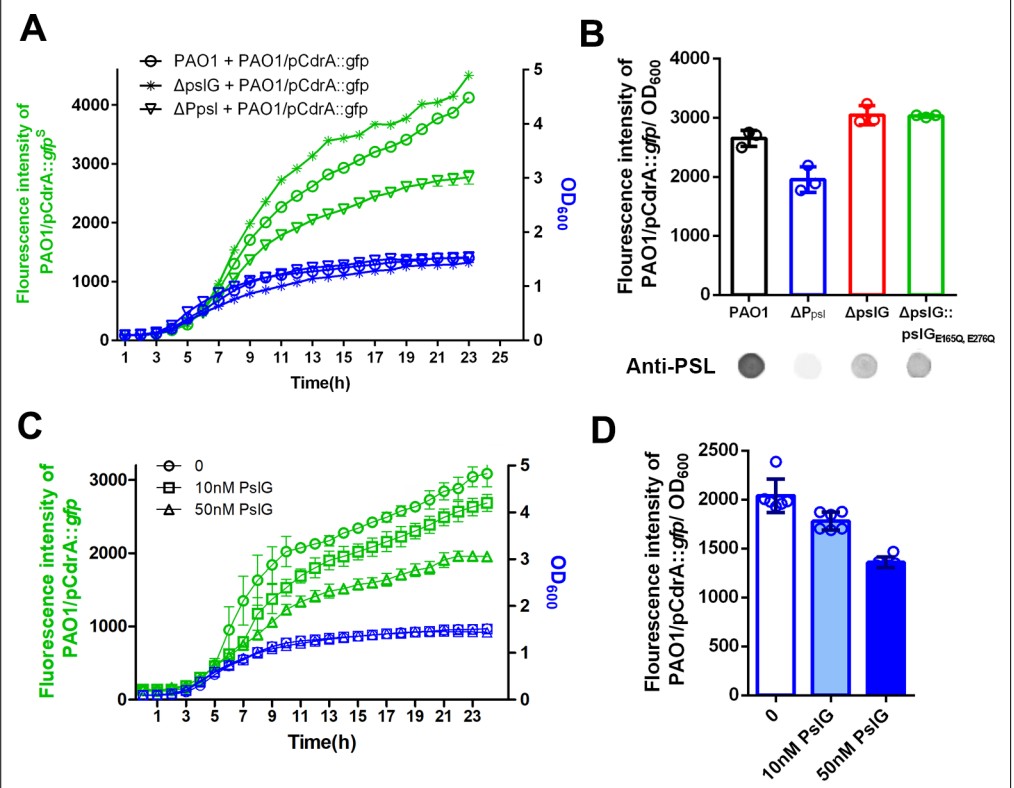

**Figure 7.** PSL produced by pslG mutant has a stronger signaling function. (A): A coculture system using pCdrA::*gfp* as a reporter plasmid to evaluate the intracellular c-di-GMP. The reporter strain (PAO1/pCdrA::*gfp*) was mixed with PSL provider strain (PAO1, ΔpslG, or ΔPpsl: a PSL-negative mutant) at a ratio of 1: 1, and the GFP fluorescence value and $OD_{600}$ in each co-culture system was recorded once per hour for 24 hr. (B): The fluorescence intensity per $OD_{600}$ of PAO1/pCdrA::*gfp* after 24 hr co-culture with PSL provider strain PAO1, ΔPpsl, ΔpslG, or ΔpslG::*pslG*$_{E165Q,E276Q}$. PSL production of each donor strain was shown under corresponding column. Statistical significances were measured using student's t-test (**, p < 0. 01 when compared to PAO1). (C): The fluorescence intensity and corresponding $OD_{600}$ of PAO1/pCdrA::*gfp* during 24 hr of PslG treatment. (D): The fluorescence intensity per $OD_{600}$ of PAO1/pCdrA::*gfp* post 24 hr of treatment with different concentration of PslG.

mechanism (*Zhao et al., 2013*), during which founder cells can be very important. The cells of two-bright cases have high c-di-GMP levels and thus can act as founder cells to promote microcolony formation. Altogether, the slight change in c-di-GMP levels of daughter cells in *pslG* mutants can be likely one of reasons to promote the formation of microcolony and long bacterial chains.

## PSL produced by *pslG* mutants has a stronger signaling function

As shown in *Figure 5*, PSL is often localized around septa in Δ*pslG* strains. Since PSL can have a signaling function by stimulating intracellular c-di-GMP production (*Irie et al., 2012*), we speculated that PSL produced from Δ*pslG* might have different signaling properties. To test the signaling functions of PSL, we set up a co-culture system, which contained a PSL donor strain and a reporter strain. PAO1 harboring the plasmid pCdrA::*gfp* was utilized as an intracellular c-di-GMP reporter strain while PAO1, ΔP$_{psl}$, Δ*pslG*, and Δ*pslG*:: *pslG*$_{E165Q,E276Q}$ strains as PSL donors respectively, in which ΔP$_{psl}$ (named as WFPA800 previously) was used as a negative control because it does not produce PSL due to deletion of the promoter of *psl* operon (*Figure 1A*). As shown in *Figure 7*, PAO1 that produces wild type level of PSL can induce a stronger GFP fluorescence signal in the reporter stain compared to ΔP$_{psl}$ (*Figure 7A,B*). Strikingly, Δ*pslG* strain and the hydrolytic activity sites mutant (Δ*pslG*:: *pslG*$_{E165Q,E276Q}$) stimulated a higher fluorescence signal than that of PAO1 (*Figure 7B*) although both of which produce less PSL (about 30% of PAO1 level, *Figure 1A*), suggesting that PSL synthesized from these two *pslG* mutants has a stronger signaling effect on stimulating the intracellular c-di-GMP production than that of wild type. We then examined the impact of exogenous PslG on the fluorescence signal of

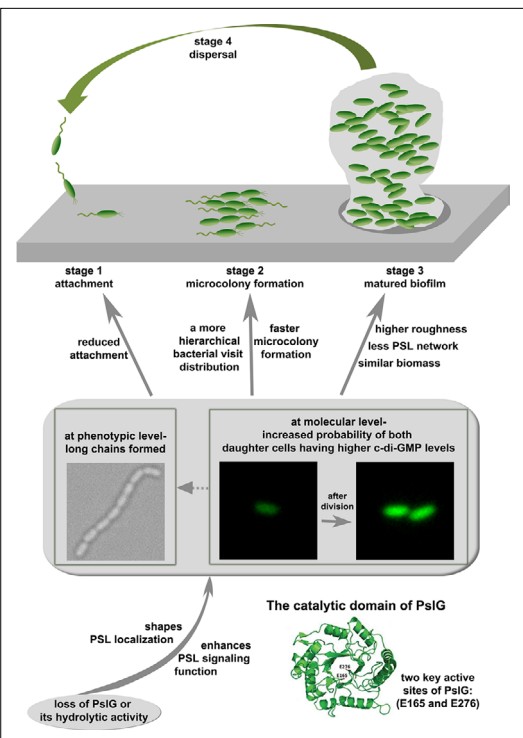

**Figure 8.** A schematic to show the effects of *pslG* in the biofilm development of *P. aeruginosa*.

PAO1/pCdrA::*gfp*. After 10 hr of PslG treatment, the fluorescence intensity of PAO1/pCdrA::*gfp* was significantly reduced compared to the non-treatment control and it had a dose-dependent manner ( *Figure 7C, D*). This further suggests that the hydrolysis of PslG might reduce the signaling property of PSL.

Taken together, our results show that lack of intracellular PslG shapes the localization of PSL on bacterial periphery, enhances the signaling function of PSL, which might further affect the c-di-GMP distribution in daughter cells during cell division, leads to changing of bacterial surface exploration, daughter cells being connected together after cell division and thus the formation of bacterial chains and microcolonies. Strikingly, all these phenotypes depend on the hydrolytic activity of PslG and PSL production, implying that PslG, as a glycosyl hydrolase, can modulate the signaling property of PSL by its hydrolytic activity to affect *P. aeruginosa* biofilm development.

## Discussion

PslG has been shown to be an efficient PSL degrader in vitro and in biofilm matrix, yet its physiologically relevant role within a bacterium or biofilm is ill-defined. In this work, by systematically studying the *pslG* knock-out mutants both at a single cell and community level, the effects of *pslG* on the bacterial physiology and surface behavior have been illustrated comprehensively. Based on our results, we propose a model for the role of *pslG* in the biofilm development of *P. aeruginosa* as the following (*Figure 8*).

The structural analysis of PslG has shown that it has a catalytic domain for the hydrolysis of PSL, and is typically considered to modify or hydrolyze the PSL polymer before PSL are secreted out of the bacterial cell (*Yu et al., 2015*; *Baker et al., 2015*). Earlier studies have shown that PslG is localized mainly on the inner membrane and a little in the periplasm (*Wu et al., 2019*; *Baker et al., 2015*). It has been proposed that PslG on membrane hydrolyzes PSL polymer during biosynthesis to help the release of PSL at right time (*Wu et al., 2019*) and the portion in periplasm is to degrade any PSL accumulated in the periplasm space (*Baker et al., 2015*). In this study, our results revealed that it is very important for *P. aeruginosa* and its communities to have a glycosyl hydrolase encoding gene, *pslG* within the polysaccharide synthesis gene cluster. Loss of PslG or its catalytic activity affects the properties and functions of PSL, including its localization and signaling. The consequence of these changes has two aspects. One is that phenotypically long chains of cells are formed. The long chains are not observed in strains with intact *pslG*. Such long chains seem not induced by the adherence on a surface as they can also be observed in liquid cultures. Rather, there are multiple lines of evidences to support that PSL is the main factor for the formation of long chains. Firstly, by adding PslG externally into cell cultures, long chains were observed to disassemble into single cells, presumably due to the hydrolysis of PSL by PslG as shown in literature (*Yu et al., 2015*; *Baker et al., 2015*; *Zhang et al., 2018*). Secondly, such long chains are not observed in P$_{BAD}$-*pslΔpslG* when PSL production is not induced (under 0% arabinose). Thirdly, the fluorescence staining experiments also show that PSL are often localized around septa of cells, where daughter cells are supposed to be disconnected after division.

The other aspect as a consequence of the change in PSL due to the loss of PslG is that, at the molecular level, the probability of both daughter cells having a higher level of c-di-GMP than that of their mother cell in a division event is increased. The increased level of c-di-GMP then would result in reduced cell motility and promote cells to transit to biofilm style (*Römling et al., 2013*). This may

also help cells to form long chains by reducing the breakage of chains due to reduced cell motility and increased PSL production.

As it is widely known that PSL plays a very important role in the biofilms of *P. aeruginosa*, then both aforementioned aspects would have an effect on the biofilm development at different stages. At the attachment stage, reduced swimming motility, which can be caused by the higher level of c-di-GMP, can contribute to the reduced surface attachment observed on microtiter surfaces. Long chains of cells may also contribute to the reduced swimming motility. However, under our experimental setup, we cannot measure the swimming of long chains in liquid cultures when they move toward to the air-liquid interface. After cells attach on a surface, both the formation of long chains and the increased level of c-di-GMP in two daughter cells will help both daughter cells to stay on the surface, and result in a more hierarchical bacterial visit distribution (*Figure 3*), which then lead to an earlier formation of microcolony. As cells continue to grow and proliferate, such differences in cell behavior between Δ*pslG* and PAO1 finally result in matured biofilms with different structures as shown in pellicles formed at the air-liquid interface, where pellicles of Δ*pslG* are rougher than those of PAO1 although their biomasses are similar. We note that the aforementioned effect of *pslG* on the biofilm development is also dependent on the synthesis of PSL.

Thus, by revealing the important roles of PslG on bacterial physiology and the biofilm development, our results further expand our understandings of PslG functions. To the best of our knowledge, the ability of a glycosyl hydrolase to affect bacterial c-di-GMP levels has not been reported until this work. It would be also very interesting to see in the future whether other glycosyl hydrolases in different bacterial species can have similar functions.

Similarly, glycosyl hydrolase has also been found to be encoded in gene clusters that are involved in the synthesis of other exopolysaccharides. For example, *algL*, a gene within the alginate synthesis operon of *P. aeruginosa*, also encodes a lyase to degrade alginate (*Franklin et al., 2011*). An early work showed that Δ*algL* in a CF isolate mucoid strain FRD1 results in cell death, due to accumulation of alginate in the periplasm (*Jain and Ohman, 2005*). Interestingly, a later work showed that Δ*algL* in a PAO1-derived mucoid strain PDO300 does not cause cell death, yet increases alginate production (*Wang et al., 2016*). A recent study further revealed that the function of AlgL in PAO1 is to clear the periplasmic space of accumulated alginate during polymer biosynthesis, while lack of AlgL might inhibit bacterial growth under certain conditions (*Gheorghita et al., 2022*). *pelA* is another example of encoding a hydrolase in the PEL synthesis operon (*Franklin et al., 2011*), which has been shown to inhibit biofilm formation as that of PslG (*Baker et al., 2016*). It would be interesting to investigate whether *algL or pelA* affects bacterial intracellular c-di-GMP levels.

In summary, in this study, we have provided a comprehensive analysis on the effect of *pslG* on the biofilm development of *P. aeruginosa*. Our results indicate that although *pslG* is not essential for synthesis of PSL, it plays an important role in regulating the proper functions of PSL, and loss of *pslG* or its hydrolytic activity results in malfunction of PSL, which then cause changes in both morphology and surface behavior of bacterial cells through PSL-mediated interactions. This work shed light on better understanding the role of PslG and would be helpful in developing new ways for biofilm control through *pslG*-based PSL regulation.

# Materials and methods

## Key resources table

| Reagent type (species) or resource | Designation | Source or reference | Identifiers | Additional information |
|---|---|---|---|---|
| Strain, strain background (*Pseudomonas aeruginosa*) | PAO1 | *Holloway, 1955* | Prototroph. | |
| Strain, strain background (*Pseudomonas aeruginosa*) | Δ*pslG* | *Wu et al., 2019* | In-frame deletion of *pslG*. | Ma's lab, strain No. P539. |
| Strain, strain background (*Pseudomonas aeruginosa*) | ΔP$_{psl}$ | *Ma et al., 2006* | PSL-negative, the promoter of *psl* operon deletion mutant. | Previous name is WFPA80. |
| Strain, strain background (*Pseudomonas aeruginosa*) | P$_{BAD}$-*psl* | *Ma et al., 2006* | PSL-inducible strain, the promoter of *psl* operon is replaced by *araC*-P$_{BAD}$. | Previous name is WFPA801 |

*Continued on next page*

*Continued*

| Reagent type (species) or resource | Designation | Source or reference | Identifiers | Additional information |
|---|---|---|---|---|
| Strain, strain background (*Pseudomonas aeruginosa*) | ΔP$_{pel}$ | *Ma et al., 2012* | PEL-negative, the promoter of *pel* operon deletion mutant. | Previous name is WFPA830. |
| Strain, strain background (*Pseudomonas aeruginosa*) | P$_{BAD}$-*pel* | *Ma et al., 2012* | PEL-inducible strain, the promoter of *pel* operon is replaced by *araC*-P$_{BAD}$. | Previous name is WFPA831. |
| Strain, strain background (*Pseudomonas aeruginosa*) | P$_{BAD}$-*psl* Δ*pslG* | This study | In-frame deletion of *pslG* in P$_{BAD}$-*psl* background. | Ma's lab, strain No. P977. |
| Strain, strain background (*Pseudomonas aeruginosa*) | Δ*pslG* ΔP$_{pel}$ | This study | The promoter of *pel* operon deletion strain in Δ*pslG* background. | Ma's lab, strain No. P1717. |
| Strain, strain background (*Pseudomonas aeruginosa*) | P$_{BAD}$-*psl* Δ*pslG* ΔP$_{pel}$ | This study | The promoter of *pel* operon deletion strain in P$_{BAD}$-*psl* Δ*pslG* background. | Ma's lab, strain No. P1711. |
| Strain, strain background (*Pseudomonas aeruginosa*) | Δ*algD* | *Whitchurch et al., 2002* | Alginate-negative, the *algD*:: *tet* deletion mutant of PAO1. | Previous name is WFPA1. |
| Strain, strain background (*Pseudomonas aeruginosa*) | Δ*pslG*::*pslG* | *Wu et al., 2019* | *pslG* was inserted into *pslG* deletion mutant at chromosome *pslG* locus. | Ma's lab, strain No. P963. |
| Strain, strain background (*Pseudomonas aeruginosa*) | Δ*pslG*::*pslG*$_{E165Q, E276Q}$ | *Wu et al., 2019* | *pslG* was replaced by the active sites mutated *pslG* (E165Q + E276 Q). | Ma's lab, strain No. P964. |
| Strain, strain background (*Pseudomonas aeruginosa*) | Δ*pslG attB*::P$_{BAD}$-*pslG* | *Wu et al., 2019* | P$_{BAD}$-*pslG* was inserted into *pslG* deletion mutant at chromosome *attB* site. | Ma's lab, strain No. P1716. |
| Strain, strain background (*Pseudomonas aeruginosa*) | P$_{BAD}$-*psl* Δ*pslG*::*pslG* | This study | *pslG* was inserted into P$_{BAD}$-*psl* Δ*pslG* strain at chromosome *pslG* locus. | Ma's lab, strain No. P967. |
| Strain, strain background (*Pseudomonas aeruginosa*) | P$_{BAD}$-*psl*Δ*pslG*::*pslG*$_{E165Q,E276Q}$ | *Wu et al., 2019* | *pslG* was replaced by the active sites mutated *pslG* (E165Q, E276Q) in P$_{BAD}$-*psl* strain. | Ma's lab, strain No. P966. |
| Strain, strain background (*Pseudomonas aeruginosa*) | P$_{BAD}$-*psl* Δ*pslG attB*::P$_{BAD}$-*pslG* | This study | P$_{BAD}$-*pslG* was inserted into P$_{BAD}$-*psl* Δ*pslG* strain at chromosome *attB* site. | Ma's lab, strain No. P1715. |
| Strain, strain background (*Pseudomonas aeruginosa*) | PAO1/pCdrA::*gfp* | *Rybtke et al., 2012* | PAO1 strain carrying plasmid pCdrA::*gfp*. Amp$^R$, Gm$^R$. | |
| Strain, strain background (*Pseudomonas aeruginosa*) | PAO1/pHERD20T | *Wu et al., 2019* | PAO1 strain carrying plasmid pHERD20T | |
| Strain, strain background (*Pseudomonas aeruginosa*) | Δ*pslG* /pHERD20T | *Wu et al., 2019* | Δ*pslG* strain carrying plasmid pHERD20T | |
| Strain, strain background (*Pseudomonas aeruginosa*) | Δ*pslG* /pG | *Yu et al., 2015* | Δ*pslG* strain carrying plasmid pHERD20T-*pslG*. Amp$^R$ | |
| Strain, strain background (*Pseudomonas aeruginosa*) | Δ*pslG* /pGDM | *Wu et al., 2019* | Δ*pslG* strain carrying plasmid pHERD20T-*pslGDM*, pHERD20T with double active sites mutated *pslG* (E165Q + E276 Q), Amp$^R$ | |
| Recombinant DNA reagent | pCdrA::*gfp* (plasmid) | *Rybtke et al., 2012* | pUCP22Not-RNase III-gfp (ASV)-T0-T1, a cyclic di-GMP level reporter consisting of the cyclic di-GMP-responsive *cdrA* promoter fused to *gfp* gene, Amp$^R$, Gm$^R$ | |
| Recombinant DNA reagent | pHERD20T (plasmid) | *Qiu et al., 2008* | pUCP20T P*lac* replaced with 1.3 kb AflIII-EcoRI fragment of *araC*-P$_{BAD}$ cassette. Amp$^R$ | |
| Recombinant DNA reagent | pG(plasmid) | *Yu et al., 2015* | pHERD20T-*pslG*. Amp$^R$ | |
| Recombinant DNA reagent | pGDM (plasmid) | *Wu et al., 2019* | pHERD20T with double active sites mutated *pslG* (E165Q + E276 Q), Amp$^R$ | |
| Recombinant DNA reagent | pEX18Gm (plasmid) | *Hoang et al., 1998* | Allelic exchange vector, Gm$^R$ | |
| Recombinant DNA reagent | pSW196 (plasmid) | *Baynham et al., 2006* | Modified from mini-CTX with pBAD30-based vector, for inserting an arabinose-inducible gene at the neutral *attB* site. Tc$^R$ | |
| Recombinant DNA reagent | pFLP2 (plasmid) | *Hoang et al., 1998* | Source of Flp recombinase, Amp$^R$ | |
| Antibody | anti-ePsl (Rabbit polyclonal) | *Byrd et al., 2009* | Exopolysaccharide Psl specific antibody. | IF(1:1667) |
| Other | TRITC-HHA | EY-lab, INC | Fluorescent labeled lectin HHA. | |

*Continued on next page*

*Continued*

| Reagent type (species) or resource | Designation | Source or reference | Identifiers | Additional information |
|---|---|---|---|---|
| Other | FITC-HHA | EY-lab, INC | Fluorescent labeled lectin HHA. | |
| Other | Syto9 | Invitrogen, Molecular probes | Green-fluorescent nucleic acid stain. | |
| Other | FM4-64 | Invitrogen, Molecular probes | Lipophilic Styryl Dye. | |

## Bacterial strains and growth conditions

All *P. aeruginosa* stains used in study were listed in the key resources table. *P. aeruginosa* stains were grown at 37 °C in LB without sodium chloride (LBNS) or Jensen's, a chemically defined media (*Jensen et al., 1980*). Biofilms of *P. aeruginosa* were grown in Jensen's medium at 30 °C. L-arabinose (Sigma) was used as inducer for genes transcribed from $P_{BAD}$ promoter in *P. aeruginosa*. Antibiotics for *P. aeruginosa* were added at the following concentrations: gentamicin 30 µg/mL; ampicillin 100 µg/mL; carbenicillin 300 µg/mL. For *Pseudomonas* selection media, irgasan at 25 µg/mL was used.

The *psl*-inducible strains $P_{BAD}$-*pslΔpslG* was constructed in accordance with $P_{BAD}$-*psl* (the promoter of *psl* operon in PAO1 was replaced by *araC*-$P_{BAD}$-*psl*) (*Ma et al., 2006*). Briefly, plasmid pMA9 (*Ma et al., 2006*) was transferred into *pslG* deletion mutant by conjugation (*Wu et al., 2019*). All deletion mutants were constructed by the similar in-frame deletion strategy (*Ma et al., 2006*). For single recombination mutant selection, LBNS plates with 30 µg/mL gentamycin and 25 µg/mL irgasan were used; for double recombination mutant selection, LBNS plates containing 10% sucrose were used. Gene insertion at the attB site of *P. aeruginosa* was performed as described previously (*Hoang et al., 2000*).

## Bacterial attachment on microtiter dish

The assay was done as described previously with modifications (*Ma et al., 2006*; *O'Toole, 2011*). Overnight culture was 1/100 diluted into Jensen's media (with or without arabinose) and incubated at 37 °C with shaking until the $OD_{600}$ reached 0.5. The 100 µL of such culture was inoculated into 96-well PVC microtiter dish (BD Falcon), and incubated at 30 °C for 30 min. Then the planktonic and loosely adherent bacteria cells were washed off by rinsing the plate in water. The remaining surface-attached cells were stained by 0.1% crystal violet, solubilized in 30% acetic acid, and finally the value of $OD_{560}$ was measured.

## Motility assay

Swimming motility assay was performed as preciously described (*Zhao et al., 2018*). Briefly, strains were grown overnight on LBNS plates. Single colony was stab-inoculated with a sterile toothpick on the surface of Jensen's plates (0.3% BD Bacto Agar). Plates were incubated upright at 37 °C overnight. Swimming zones were measured accordingly. For the twitching motility assay, the strains were stab inoculated with a sterile toothpick into the bottom of thin Jensen's plates, cultivated at 30 °C for 2–3 days, and the twitching motility zones were visualized at the agar plate interface (*Wang et al., 2013*).

## PSL dot-blotting

Strains were incubated in Jensen's medium with shaking at 30 °C for 24 hr. Cells of an $OD_{600}$ of 4 were collected by centrifugation to extract crude bacterial surface-bound exopolysaccharides. Pellet was re-suspended in 100 µL of 0.5 M EDTA, and boiled at 100 °C for 5 min. After centrifugation at 13,000 g for 10 min, the supernatant fraction was treated with 0.5 mg/mL proteinase K at 60 °C for 1 hr and proteinase K was then inactivated at 80 °C for 30 min. PSL immunoblotting was performed as previously described using PSL antibody (*Byrd et al., 2009*). ImageJ software was used to quantify the immunoblot data. The protein concentration of each sample culture was measured by a BCA protein assay kit (Thermo) to ensure the same amount of cell lysate was used in each experiment.

## Flow cell assembly, sterilization, and washing of the system

Flow cells made of polycarbonate were purchased from the Department of Systems Biology, Technical University of Denmark. Each flow cell has three identical rectangle channels ($40 \times 4 \times 1$ mm$^3$) and was assembled by attaching a cover glass as substratum as previously described (*Sternberg and Tolker-Nielsen, 2006*). The assembled flow cell was connected to a syringe through a 0.22 μm filter (Millipore) using silicon tubing. Then the whole system was sterilized overnight with 3% $H_2O_2$ at 3 mL/hr using a syringe pump (Harvard Apparatus). After sterilization, autoclaved, deionized water was used to wash the whole system overnight. Before inoculation of bacteria into the flow cell, the system was flushed for 5 min at a flow rate of 30 mL/hr by Jensen's medium using a syringe pump (Harvard Apparatus). Then the medium flow was stopped and 1 mL of a diluted bacteria culture (OD$_{600}$ ~ 0.01) were injected directly into the channel of the flow cell using a 1 mL syringe equipped with a needle. A 5-min incubation period was allowed after inoculation to let cells attaching to the surface, which was then followed by a medium flow with a large flow rate of 30 mL/hr for 5 min to wash out floating cells. After that the flow rate was set to 3 mL/hr, and image recording was started. In this work, the flow cell experiments were conducted at 30 °C.

## Biofilms and image acquisition

Pellicles (air-liquid interface biofilms) were grown in glass chambers (Chambered # 1.5 German Coverglass System, Nunc) with a glass coverslip at the bottom of each chamber as described previously (*Wang et al., 2013*). 1/100 dilution of a saturated (overnight) culture in Jensen's media for *P. aeruginosa* was inoculated into the chamber, and incubated at 30 °C for 24 hr. The PSL was stained with lectin TRITC-HHA (EY lab, INC) at 100 μg/mL for 2 hr in the dark. Then bacteria were strained with SYTO9 (5 μM final concentration, Molecular Probes, Invitrogen) for 15 min. Fluorescent images were obtained using a FV1000 CLSM (Olympus, Japan). The excitation/emission parameters for TRITC-HHA and SYTO9 were 554 nm/570 nm and 480 nm/500 nm, respectively. CLSM-captured images were analyzed using COMSTAT software (*Heydorn et al., 2000*).

For flow cell experiments, the flow was stopped before staining. PSL was stained with lectin FITC-HHA (EY lab, INC) at 100 μg/mL for 20 min in the dark, and then the flow was running for a short time to flush out the non-binding dye. Subsequently, bacteria were tagged by Gfp or stained by cell membrane stain FM4-64 (10 μM final concentration, Molecular Probes) for 2 min in the dark (without flow). Next, the flow was resumed to flush out the dye and ready for examination under microscope. Images were captured using an EMCCD camera (Andor iXon) on a Leica DMi8 microscope equipped with Zero Drift autofocus system. The image size is 66.5 μm × 66.5 μm (1,024 × 1,024 pixels). The images were recorded with a 100 × oil objective (plus 2 × magnifier).

## Detection of PSL signaling function on stimulating bacterial intracellular C-di-GMP

The c-di-GMP levels were determined using pCdrA::*gfp* as a reporter as described previously (*Rybtke et al., 2012*). The growth curve and green fluorescent signal of PAO1 /pCdrA::*gfp* were measured via recording the OD$_{600}$ values and the corresponding GFP fluorescence (Ex/Em 488/520) by a Synergy H4 hybrid reader (BioTek). The promoter activity was calibrated as the relative fluorescence divided by the OD$_{600}$. In co-culture system, PAO1 harboring plasmid pCdrA::*gfp* was the reporter strain to indicate the level of intracellular c-di-GMP. PAO1, P$_{BAD}$-*psl*, Δ*pslG*, and Δ*pslG*::*pslG*$_{E165Q,E276Q}$ strains were PSL donor strains, respectively. In PslG treatment assay, PslG was added into cultures when inoculating PAO1 /pCdrA::*gfp*, the OD$_{600}$ values and the corresponding GFP fluorescence were tracked for 24 hr.

## Single-cell tracking image analysis

Images were processed and analyzed in the same way as described in reference (*Zhang et al., 2018*). Simply, 16-bit greyscale images were first converted to binary images for the detection of bacteria with a standard image processing algorithm. Geometry information of cells such as center position, size and aspect ratio etc. were then collected. Bacterial trajectories were obtained by connecting cell positions in all frames of a time series, from which bacterial motion can be measured and analyzed. Specifically, the twitching speed of each tracked cell at frame n was calculated by the displacement of the cell between n$^{th}$ and (n + 1)$^{th}$ frames divided by the corresponding time interval.

For quantitatively comparing the fluorescence intensity of cells containing pCdrA::*gfp* reporter between mother cell and daughter cells, first the fluorescence intensity of each cell $I$ was measured by the averaged fluorescence intensity value within the area enclosed by the cell envelope. The mother cell was measured when it irreversibly attached to the surface (typically 40~50 min before the division completed), and the daughter cells were measured right after the division (i.e. the two daughter cells are completely separated. In practice, the daughter cells were measured within 10 min right after the division completed due to the 10-min time interval for the fluorescent image recording). Then the ratio of fluorescence intensity between each daughter cell ($I_{dau}$) and its mother cell ($I_{mot}$) was calculated, $\gamma = I_{dau}/I_{mot}$. The relative standard deviation was estimated by $\frac{SD_\gamma}{\gamma} = \sqrt{(\frac{SD_{I_{dau}}}{I_{dau}})^2 + (\frac{SD_{I_{mot}}}{I_{mot}})^2}$. Here, $SD_\gamma$ refers to standard deviation of $\gamma$. Similar for $SD_{I_{dau}}$ and $SD_{I_{mot}}$. We define a daughter cell to be fluorescent brighter than its mother cell if $\gamma > 1 + (\overline{\frac{SD_\gamma}{\gamma}})$, here $(\overline{\frac{SD_\gamma}{\gamma}})$ is the averaged value of the relative standard deviation for all analyzed division events.

A cluster is an aggregation of multiple cells. We used a minimum distance criterion to judge whether a cell belonged to a cluster or not. If the minimum distance between any point of the scrutinized cell body and any point of any cell body of the cluster, is smaller than 0.5 μm (i.e. about one width of a bacterial cell), then the scrutinized cell is considered to belong to the cluster, otherwise not.

## Acknowledgements

This work is supported by the National Key R&D Program of China (2018YFA0902102, 2021YFA0909500, 2019YFC1804104, and 2019YFA0905501), the National Natural Science Foundation of China (91951204, 21621004, 32070033), and Research on basic science and technology of the strategic reserve fund projects of PetroChina Company Limited (2020D-5008-01). The funders had no role in the study design, data collection and interpretation, or the decision to submit the work for publication.

## Additional information

### Funding

| Funder | Grant reference number | Author |
| --- | --- | --- |
| National Key Research and Development Program of China | 2018YFA0902102 | Kun Zhao |
| National Natural Science Foundation of China | 91951204 | Luyan Ma |
| National Natural Science Foundation of China | 32070033 | Di Wang |
| National Natural Science Foundation of China | 21621004 | Kun Zhao |
| National Key Research and Development Program of China | 2019YFA0905501 | Di Wang |
| National Key Research and Development Program of China | 2019YFC1804104 | Luyan Ma |
| National Key Research and Development Program of China | 2021YFA0909500 | Luyan Ma |
| PetroChina Company Limited | 2020D-5008-01 | Huijun Wu |

The funders had no role in study design, data collection and interpretation, or the decision to submit the work for publication.

## Author contributions
Jingchao Zhang, Investigation, Writing – original draft, Writing - review and editing; Huijun Wu, Investigation, Writing – original draft; Di Wang, Investigation, Project administration, Writing – original draft, Writing - review and editing; Lanxin Wang, Yifan Cui, Chenxi Zhang, Investigation; Kun Zhao, Conceptualization, Data curation, Formal analysis, Funding acquisition, Methodology, Project administration, Supervision, Validation, Writing – original draft, Writing - review and editing; Luyan Ma, Conceptualization, Data curation, Formal analysis, Funding acquisition, Methodology, Project administration, Resources, Supervision, Validation, Visualization, Writing – original draft, Writing - review and editing

## Author ORCIDs
Kun Zhao ⓘ http://orcid.org/0000-0003-3928-1981
Luyan Ma ⓘ http://orcid.org/0000-0002-3837-6682

## Decision letter and Author response
Decision letter https://doi.org/10.7554/eLife.72778.sa1
Author response https://doi.org/10.7554/eLife.72778.sa2

# Additional files

## Supplementary files
• Transparent reporting form

## Data availability
All data generated or analysed during this study are included in the manuscript and supporting file. Source Data files have been provided for Figures 1, 2, 3, 4, 6 and 7.

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
