## [Editor Report]

One of the major findings related to the production of long chain polysaccharide polymers by bacteria is why there often is a hydrolytic enzyme encoded within the gene cluster encoding the biosynthetic proteins. At a basic level, these hydrolases cleave the chains to prevent toxic accumulation of intracellular polysaccharides, but it has now been discovered that a glycosyl hydrolase acting on the *Pseudomonas aeruginosa* PSL polysaccharide affects other important responses of this Gram-negative bacterium. The PslG hydrolase regulated cell length, PSL cell surface localization and signaling via cyclic-di-GMP, overall controlling accumulation of the cells into a protective biofilm.

---

## [Decision Letter]

**Decision letter after peer review:**

[Editors’ note: the authors submitted for reconsideration following the decision after peer review. What follows is the decision letter after the first round of review.]

Thank you for submitting the paper "Intracellular glycosyl hydrolase PslG shapes bacterial cell fate, signaling, and the biofilm development of *Pseudomonas aeruginosa*" for consideration by *eLife*. Your article has been reviewed by 3 peer reviewers, and the evaluation has been overseen by a Reviewing Editor and a Senior Editor. The following individuals involved in review of your submission have agreed to reveal their identity: Joe Jonathan Harrison (Reviewer #2).

Comments to the Authors:

We are sorry to say that, after consultation with the reviewers, we have decided that this work will not be considered further for publication by *eLife*.

While there was clear interest in the overall subject and the direction of the experimental approaches, the reviewers found multiple deficiencies in the overall experimental design that diminished the strength of the conclusions based on the data presented. Thus, the ability to complete numerous experiments needed to strengthen the data within 2 months is highly unlikely and serves as the basis for the decision. In particular the lack of trans-complemented strains or use of strains with point mutations in PslG raised issues about pleiotropic effects and specificity. Lack of building on prior work in the field and citing prior results on the alginate lyase, AlgL, which led to a similar phenotype, diminishes the claims to novelty and primacy. Also, the impact of two other surface polysaccharides, Pel and alginate was not explored. Most *Pseudomonas aeruginosa* strains make alginate at a low level, but PAO1 does not do so under aerobic conditions in vitro, so use of this strain as the major one to test the hypotheses and draw conclusions could represent findings not generally applicable to *P. aeruginosa*. Additional concerns about the interpretation of the Raman spectra and weaknesses in the data that support the conclusions regarding the basis for the effect of loss of PslG on the cellular structures further diminish the impact of the findings.

*Reviewer #1:*

The manuscript primarily presents phenotypic results to shape their story and conclusions. Given the importance of both Psl and Pel to exopolysaccharide-dependent behaviors in *P. aeruginosa* PAO1 (and many other strains), the potential importance and contribution of Pel is essentially ignored here. Because both Pel and Psl can generally influence several of the phenotypes explored in this manuscript, the authors' desire to draw specific conclusions from the effect of PslG cannot be separated from potential contributions of Pel polysaccharide in these experiments. Another difficulty in assessing the results and conclusions lies in the fact that some experiments utilize only the WFPA801ΔpslG or ΔpslG strains, which are inherently different (Figure 3 shows ΔpslG with PAO1 as a control while Figure 4 shows WFPA801ΔpslG with PAO1 as a control). Thus, there are two parallel lines of evidence that do not fully converge. In addition, as the pslG mutation is not complemented in any set of experiments, the specificity of pslG to the effects detailed here is not definitely demonstrated. There may be much to be learned by probing an arabinose-inducible pslG construct as the authors have done with overall Psl production.

The most compelling data in the manuscript involves the impact of pslG upon divided cells. Both the c-di-GMP reporter and cell-chain data point to a role of PslG in disposition of progeny. These results included in Figures 4, 6, and 7 are fascinating but require further probing to actually explain the role of PslG.

A deeper interrogation of cell division may uncover a direct connection between PslG and the cell cycle. Several other types of experiments may be useful to discern this connection.

The manuscript misses the opportunity to reference and build upon recent work by the Wozniak and Parsek labs on Pel and Psl (from just the last 6 years).

Line 116. From all of the descriptions here, the methods, and the figure 1 legend, it is not clear what has been quantified in Figure 1A as the measure of Psl.

Line 131. Even though the WFPA801ΔpslG strain shows a "statistically significant" decrease in swim zone diameter in comparison to WT or un-induced conditions, the difference is minor. Certainly, this strain exhibits flagellar motility. Given the small difference in flagellar motility, which is used here as an indirect assessment of flagellar function that may apply to surface attachment, it would be more prudent to conclude "…the defect of ΔpslG on initial attachment MAY BE the result of multiple contributing factors."

Line 173. "…by changing the surface exploration at early stages of biofilm formation…" I disagree this is what the data show here and suggest this contradicts the data discussed lines 157-164 in the same paragraph where surface coverage differences are not apparent between pslG and PAO1 at N<10,000. Indeed, it would be the impact of pslG upon the 10,000-100,000 range of visits that begs for further discernment. Or what is the working definition of "early stages" here?

Line 228. The Raman spectra of the three strains clearly show many differences in presence and intensity of numerous peaks between 800-1800 cm-1. In the absence of more specific and supportive references and other controls, it is not clear that the 865 cm-1 peak is the definitive feature between these three spectra nor is it clear this peak allows for the specific discernment of a 1,4-glycosidic linkage polysaccharide. Lastly, it is not clear how the authors can even discern Psl from Pel polysaccharide by Raman given the data presented here. There may be no "structural change in Psl" observed here but rather a shift from Psl to Pel.

Line 238. These results are difficult to interpret. If the ΔpslG strain makes less Psl, then how is the chain-forming phenotype dependent upon increased Psl production in the WFPA801ΔpslG strain?

*Reviewer #2:Pseudomonas aeruginosa* is an opportunistic pathogen that is one of the leading causes of bacterial infection worldwide. The ability of this bacterium to form biofilms, which are aggregates of these microorganisms encased in a protective layer of polymers, is key for its ability to survive in different environments and cause disease. In this report, the role of a glycoside hydrolase enzyme, PslG, which is co-produced with more than a dozen other enzymes used by the bacteria to build their protective biofilm polymeric matrix, is investigated. The authors provide evidence that this enzyme determines cell fate in the early stages of *P. aeruginosa* biofilm formation by enabling cell detachment post-division. Understanding the natural functions of these enzymes is important because they comprise an emerging class of experimental therapeutics that target biofilms.The data presented by the authors in the manuscript are intriguing. However, the paper needs to be more carefully written because the impact will be tempered by the presentation. Understanding the natural role of glycoside hydrolases in biofilm development is a fundamental problem for the field that microbiologists have been pondering for more than a decade. Most of the conclusions are supported by data, but findings are not communicated as effectively as they could be, which detracts from what otherwise is a very nice dataset. There are some controls that are missing as well.

Key constructive criticisms

The authors should complete a complementation analysis for pslG in all their assays. As the authors note, the pslG gene is in an operon and so polar effects are a concern, and because pslG mutations appear pleiotropic, ensuring that phenotypes do not result from second site mutations elsewhere in the chromosome of engineered strains seems especially important.

To ensure that the hydrolase function of PslG is responsible for the observed phenotypes, it would be prudent to repeat some key assays in which the active site residues of PslG have been disrupted by site-directed mutagenesis.

Line 110-121. It is not clear what the authors are measuring and reporting in Figure 1A. In the text, the authors make statements about the quantities of ePSL produced by the different strains. However, the data in Figure 1 has no units and it is not clear whether the authors are measuring ePSL directly, or if the data represents something else. Could the authors quantify biofilm formation shown in the pictures below Figure 1A? Also, it would be highly beneficial if a semi-quantitation of the dot blots shown was provided too. Corresponding text should be added to the figure legend to succinctly explain the data presented in the figure.

Line 134-135. Could the authors better explain how the data led to their conclusion that PslG modulates surface attachment through multiple mechanisms? It is not clear from the text why the authors conclude that changes in motility are causally linked to defects in bacterial surface attachment, because the flagellum can be dispensable for biofilm formation depending on the assay. Also, the evidence here indicates that pslG mutations are pleiotropic.

Lines 228-234 and Figure 5 C. Caution needs to be applied using Raman spectroscopy on whole biofilms because the composition of biofilms is complex. How can the authors be certain that changes at 865 cm-1 correlate specifically with a glycosidic link in ePSL? There has been a demonstration that deletion of pslG affects intracellular c-di-GMP which influences many things in physiology including multiple components of the *P. aeruginosa* biofilm matrix. The authors would need to purify PSL from the WFPA801 and WFPA801 pslG strains and analyze that material to make these conclusions with confidence.

Lines 212-227 and Figure 5A and B. Why isn't data for the pslG mutant also included in Figure 5A?

Lines 276-277. In this case, "data not shown" is not acceptable because the WFPA801 and WFPA801 pslG are engineered to have artificially high levels of PSL production. The data that are not shown correspond to key information corroborating physiological relevance to the wild type background. Please add these data to the manuscript.

*Reviewer #3:*

In this manuscript, the authors investigated the role of PslG, a *Pseudomonas aeruginosa* glycosyl hydrolase, in biofilm formation and cell physiology of a *P. aeruginosa*. PslG is encoded on the 15 gene psl operon, and a previous study indicated that deletion of pslG caused reduced ePSL (extracellular PSL polysaccharide) production. However, in the previous study, the pslG deletion may have been polar on other downstream psl genes and may have affected ePSL production. Therefore, in this study the investigators constructed a pslG deletion in a strain where the entire psl operon is controlled by the arabinose-inducible promoter. In this way, ePSL is produced but the operon lacks pslG. The investigators showed that in this strain, ePSL is produced at levels similar to the wild-type strain. However, the pslG deletion strain had several phenotypic defects that suggested a functional role of the glycosyl hydrolase. Among the defects was a reduction in biofilm formation, even when ePSL was produced, and a reduction in swimming motility. The investigators also used microscopic tracking of the cells, and noticed a reduction in surface area covered, likely due to clumping of the cells. The clumping of the cells was due to their formation of cell chains, which were not observed in the wild-type strain or in the new strain, when ePSL production was not induced with arabinose. Interestingly, the investigators noted that in the cells that were unable to separate following division (cell chains) had ePSL accumulated at the boarders between the cells. The chains could be released by adding exogenous PslG enzyme to the cultures. The investigators suggested that in the absence of PslG, the ePSL may accumulate at the cell boarders, holding the daughter cells together. In addition to these phenotypes, the investigators also noticed a difference in the production of the cyclic-di-GMP signaling molecule in the pslG mutant strain. Since c-di-GMP is an important signaling molecule for biofilm production, the investigators put forth a model for the role of PslG in c-di_GMP signaling in the development of *P. aeruginosa* biofilms.

Overall, I think this is a thorough study and the results are well presented. A weakness of this study (in my opinion) is that while adequately exploring the phenotypes of the PslG mutant in an ePSL producing strain, a functional role for PslG in polysaccharide degradation during ePSL biosynthesis is not adequately developed. The model in Figure 8 (and the abstract) implies that the role of PslG is through c-di-GMP signaling molecule. However, the data in Figure 4B and Figure 5B show only a minor effect of the pslG deletion on c-di-GMP signaling. In Figure 4B, most cells are classified as "none-bright" in both the experimental and control strain. Only a minor percentage of cells are classified as "two-bright". The difference in "two-bright" between the control and experimental in Figure 4B, although statistically different, is not a high percentage of the total cells. Figure 5A shows a moderate difference in c-di-GMP production as indicated by a reporter gene. However, Figure 5A does not include a growth curve, so the difference seen in fluorescence could be due to a difference in cell numbers/growth. This would not be surprising, given the many physiological defects of the pslG mutant. (a growth curve needs to be shown to determine if the fluorescence difference is due to cell growth differences). The model in Figure 8 implies that c-di-GMP signaling (and the "two-bright" daughter cells) is the primary signaling event for biofilm formation in the pslG mutant. In my opinion, this is overstated, given the minor effect of the pslG mutation on c-di-GMP production. As far as I know, PslG does not have a c-di-GMP binding domain, so it is difficult to draw a molecular link between the pslG mutant and c-di-GFP. The slight increase in c-di-GMP in the mutant cells may be due to an indirect effect of an overall stress response in the mutant cell, and not a direct signaling mechanism.

I recommend that the authors read and cite the publication, "Role of an alginate lyase for alginate transport in mucoid *Pseudomonas aeruginosa*" Infect Immun. 2005 Oct;73(10):6429-36.doi: 10.1128/IAI.73.10.6429-6436.2005.

In the discussion, the authors state, "To the best of our knowledge, the ability of a

glycosyl hydrolase to affect bacterial phenotype and c-di-GMP levels has not been

reported until this work. It would be also very interesting to see in the future whether

other glycosyl hydrolases in different bacterial species can have similar functions."

In the paper above, the authors characterized another glycosyl hydrolase from *Pseudomonas aeruginosa*, that degrades alginate, and is encoded on the alginate biosynthetic operon (so, a very similar model system). In that study, the authors found that deletion of algL in an alginate producing strain resulted in cell death, due to accumulation of alginate in the periplasm. There is a precedence for these types of studies. The mechanism for PslG may be different from AlgL (and the PslG deletion obviously doesn't kill the bacteria), but the AlgL study could provide a hypothesis on the molecular role of PslG. In addition, both AlgL and PslG reside in the periplasm (assuming the model in reference: Frontiers in Microbiology 2:167 is correct). The authors of the current study only mention the periplasmic location for PslG in the Discussion section. The implication in the present study is that PslG is secreted. If PslG is secreted, there should be evidence for that (or a citation that shows that it is secreted). The localization of this enzyme could be crucial in understanding its function.

[Editors’ note: further revisions were suggested prior to acceptance, as described below.]

Thank you for resubmitting your work entitled "Intracellular glycosyl hydrolase PslG shapes bacterial cell fate, signaling, and the biofilm development of *Pseudomonas aeruginosa*" for further consideration by *eLife*. Your revised article has been evaluated by Wendy Garrett (Senior Editor) and a Reviewing Editor.

The manuscript has been improved but there are some remaining issues that need to be addressed, as outlined below:

*Reviewer #1:*

The manuscript seeks to delineate the role of pslG, a glycosyl hydrolase encoded within an operon that enables the production of the Psl polysaccharide of the bacterium *Pseudomonas aeruginosa*. The results in the manuscript show numerous effects using a combination of pslG mutants and over-expression strains upon *P. aeruginosa* behavior important to colonization and biofilm development. This adds to our understanding of the multi-functional enzymes involved in regulating polysaccharide production and the overall behavior of *P. aeruginosa*.

The manuscript primarily presents phenotypic results to shape their story and conclusions. Given the importance of both Psl and Pel to exopolysaccharide-dependent behaviors in *P. aeruginosa* PAO1 (and many other strains), the potential importance and contribution of Pel remains essentially ignored in this manuscript. Because both Pel and Psl can generally influence several of the phenotypes explored in this manuscript, the authors' desire to draw specific conclusions from the effect of PslG cannot be separated from potential contributions of Pel polysaccharide in these experiments.

The story is certainly complicated as study of pslG mutants does affect levels of Psl polysaccharide. The most compelling data in the manuscript involves the impact of pslG upon divided cells. Truly fascinating. However, it is not clear that the authors demonstrate the importance of PslG to "normal" cell division and/or how Psl polysaccharide might contribute to cell division in the wild-type. There is no basis in the literature to suggest that pslG mutations would be common in select environments.

Several new lines of experimental evidence have been added to the manuscript in this revised form and these changes are a great collective improvement-but the true role(s) and effect of PslG are still not clear from the evidence presented.

There are several sentences where the English syntax and grammar require editing.

Line 159. There does not appear to be fewer bacterial cells in the pslG biofilms-and no quantification is given to support this statement.

Line 178. "…a more hierarchical bacterial visit distribution for pslG…" It is not clear how this shows hierarchical visitors or a broad range of visit frequencies. How is this description evident based upon what is shown in Figure 3B?

Line 184. There appear to be 27-28 cells in the PAO1 cluster detailed in Figure 3D.

Line 188. "…microcolonies of ΔpslG are more compact". This representation of the data in Figure 3F does not necessarily support this statement. Is this a projection view showing a 2D image of the thicker 3D biofilms as described in Figure 2? How was this quantified?

Line 221. How were these twitching speed experiments performed? These are not described in the supplemental Figure legend or the methods.

Line 240. "…were observed most frequently in either ΔpslG or PBAD-pslΔpslG strain…" The purpose of this statement is not clear. These are the only strains shown in Figure 4. Is this a comparison?

Line 241. What is the support for the statement "…yet some can also be broken by bending of chains"? This is from separate experimental evidence? Or just an observation?

Line 246. This Figure 5 data is the most intriguing of the manuscript.

Line 309. "…which might enhance their stay on the surface, reduce their surface motility, and promote the microcolony formation". Given the small variation in the data shown in Figure 6B, support for this statement really requires specific testing. Further, not all previous data in the manuscript support it. The text in an earlier section (starting line 213) effectively concludes that there is no quantifiable motility difference between these strains.

Line 497. The Psl staining with TRITC-HHA is specific to Psl? Is there binding to Pel or other polysaccharides?

Line 770. The Figure 1 title is not the best description of what is shown here. No "increasing" levels of Psl are shown. There are two. The data show adding of Psl into a pslG background.

Line 821. It is not stated anywhere in the legend or methods from how many frames the data analyzed for Figure 3E were obtained. There are error bars, but it is not clear what these represent.

Line 831. The criteria for "microcolony" is not entirely clear. Some of these cells clearly have different spacing but are outlined with dashed lines.

*Reviewer #2:*

The manuscript by Zhang and colleagues carefully describes the phenotypes of pslG mutants of *P. aeruginosa*. PslG encodes a glyocoside hydrolase. While the biochemistry of this enzyme has been understood for many years, its physiologically relevant role in *P. aeruginosa* biofilm formation has remained ill-defined.

The experiments in the manuscript have been meticulously executed. There are controls and complementation analyses that provide confidence in the results obtained. The technical proficiency with microscopy is commendable. However, while the authors provide a data-rich manuscript, an understanding of the consequences of PslG expression appears lacking beyond phenotyping.

Perhaps this criticism is most pertinent to the observed changes in the power-law distribution for bacteria during the earliest stages of biofilm formation for pslG mutants. Interpretation/experimentation is absent that connects these observations to social biology. Such connections, which are front and center in prior work published by one of the co-authors (Dr. Khun Zhao), could help to explain conservation of glycoside hydrolases among synthase-dependent exopolysaccharide secretion systems like the Psl synthase. For example, beyond the careful phenotyping presented in this paper, co-culture of Pbad-Psl with Pbad-Psl-PslG strains, or perhaps wild type and pslG strains, that have been uniquely labelled with fluorescent proteins and tracked using Dr. Kun Zhao's elegant single-cell methods could directly demonstrate fitness changes for pslG mutants in surface exploration or colonization relative to wild type. In principle, wouldn't such a fitness cost provide an explanation for PslG function that is rooted in social evolutionary theory? Perhaps there are some trade-offs that aren't yet apparent. The link to c-di-GMP signaling provides some molecular insight even the sensory perception and signal transduction pathway is not yet fully known. Such analyses could take the work assembled here to the next level with little additional experimental effort, and as such, strikes me as a missed opportunity to provide significant, additional understanding of some really nice data.

---

## [Author Response]

[Editors’ note: The authors appealed the original decision. What follows is the authors’ response to the first round of review.]

Comments to the Authors:We are sorry to say that, after consultation with the reviewers, we have decided that this work will not be considered further for publication by eLife.While there was clear interest in the overall subject and the direction of the experimental approaches, the reviewers found multiple deficiencies in the overall experimental design that diminished the strength of the conclusions based on the data presented. Thus, the ability to complete numerous experiments needed to strengthen the data within 2 months is highly unlikely and serves as the basis for the decision. In particular the lack of trans-complemented strains or use of strains with point mutations in PslG raised issues about pleiotropic effects and specificity.

We have collected some complementation data. Moreover, since we have trans-complemented strains and point mutations in PslG on hand, it would not take very long to perform more experiments if it is necessary.

Lack of building on prior work in the field and citing prior results on the alginate lyase, AlgL, which led to a similar phenotype, diminishes the claims to novelty and primacy.

We apologize for missing this paper. We will certainly add this reference and its associated information in the introduction. As pointed by the Reviewer 3, lacking of AlgL leads to cell death. Thus, the phenotype of AlgL deletion is certainly different from what we have found in PslG mutant.

Also, the impact of two other surface polysaccharides, Pel and alginate was not explored. Most *Pseudomonas aeruginosa* strains make alginate at a low level, but PAO1 does not do so under aerobic conditions in vitro, so use of this strain as the major one to test the hypotheses and draw conclusions could represent findings not generally applicable to *P. aeruginosa*.

We will perform more experiments involving strains that lack of Pel and alginate production to explore whether Pel and alginate have any contribution on the phenotype of PslG mutant. Fortunately, we have also Pel-negative strain and alginate-negative strain on hand too. So, it will not take much time to construct strains that we need for such experiments.

In addition, regarding on the alginate production of our lab PAO1 strain, in some of our earlier studies, we have measured the alginate production of our PAO1 under our lab aerobic growth conditions, and the results show that the production of alginate is relatively low, only 0.3% of PAO1-drevied mucoid strain (Ma, L.*, Wang, J., Wang, S., Anderson E.M., Lam J.S., Parsek, M.R., Wozniak, D.J. 2012, Synthesis of multiple *Pseudomonas aeruginosa* biofilm matrix exopolysaccharides is post-transcriptionally regulated. *Environ. Microbiol*. 14 (8):1995-2005). We will add corresponding discussion in the revision

Additional concerns about the interpretation of the Raman spectra and weaknesses in the data that support the conclusions regarding the basis for the effect of loss of PslG on the cellular structures further diminish the impact of the findings.

We will perform more controls to confirm Raman spectra of PslG mutant. Right now we have already Pel-negative strain, PBAD-Pel strain, and PAO1 algD deletion strain on hand, which can be used as the controls. In addition, we will construct an in-frame deletion mutant that has *pel* and *algD* deletion in δ PslG background in order to further diminish the possible impact of Pel and alginate on the Raman spectra. Since the plasmids for the construction of mutants are ready, we expect the construction of new mutants as well as new controls to be finished within a couple of months.

Reviewer #1:The manuscript primarily presents phenotypic results to shape their story and conclusions. Given the importance of both Psl and Pel to exopolysaccharide-dependent behaviors in *P. aeruginosa* PAO1 (and many other strains), the potential importance and contribution of Pel is essentially ignored here. Because both Pel and Psl can generally influence several of the phenotypes explored in this manuscript, the authors' desire to draw specific conclusions from the effect of PslG cannot be separated from potential contributions of Pel polysaccharide in these experiments.

We agree with Reviewer #1 that both PSL and PEL are important to exopolysaccharide-dependent behavior of PAO1. In the revised manuscript, we have added more text about PEL and alginate in the introduction part (See L60-L66).

Based on literature results (for example, Colvin et al., 2012, Environmental Microbiology), PAO1 is a PSL-dominant strain, whose biofilm formation is mainly dependent on PSL. Thus, we speculate that PEL may have little contribution to the phenotypes of *pslG* mutants observed in this work. Following suggestions of reviewers, we have added the results of PEL-negative strain as well as *pslG* deletion in PEL-negative background in the revised manuscript (see Figure 3—figure supplement 3 and Figure 5—figure supplement 3). As expected, all phenotypes of *pslG* mutants are still observed in PEL-negative strains, indicating that the results of this study are not relied on PEL expression.

Another difficulty in assessing the results and conclusions lies in the fact that some experiments utilize only the WFPA801ΔpslG or ΔpslG strains, which are inherently different (Figure 3 shows ΔpslG with PAO1 as a control while Figure 4 shows WFPA801ΔpslG with PAO1 as a control). Thus, there are two parallel lines of evidence that do not fully converge.

Thanks for the comment. In the revised manuscript, we have added more datasets including those for Δ*pslG* and WFPA801Δ*pslG* strain (WFPA801 is referred as P_BAD_-psl in the revised manuscript as Reviewer #2 suggested), which are shown in the revised figures (Figure 1, Figure 3, Figure 3—figure supplement 1, Figure 3—figure supplement 2, Figure 3—figure supplement 3, Figure 5—figure supplement 2, Figure 5—figure supplement 3, Figure 6, Figure 6—figure supplement 1, Figure 7).

In addition, as the pslG mutation is not complemented in any set of experiments, the specificity of pslG to the effects detailed here is not definitely demonstrated. There may be much to be learned by probing an arabinose-inducible pslG construct as the authors have done with overall Psl production.

We thank the Reviewer for the comment. Following reviewers’ suggestions, we have performed more experiments, including those for complementation analysis of *pslG* by constructing complemented strains. We have tested not only Δ*pslG:*:*pslG* and P_BAD_-*psl*∆*pslG*::*pslG* strains, where *pslG* was inserted into *pslG* deletion mutant at chromosome *pslG* locus, but also Δ*pslG attB*::P_BAD_-*pslG* and P_BAD_-*psl* Δ*pslG attB*::P_BAD_-*pslG* strains, where P_BAD_-*pslG* was inserted into *pslG* deletion mutant at chromosome *attB* site. The results show that they all can complement the phenotypes observed in this work. In the revised manuscript, we have added these results in Figure 1, Figure 3—figure supplement 1, and Figure 5—figure supplement 2.

In addition, results in an earlier work (Wu et al. 2019, *MicrobiologyOpen*) also showed that arabinose-inducible *pslG* in plasmid vector pHerd20T can complement PSL production of the *pslG* deletion mutant (same strain used in this study).

The most compelling data in the manuscript involves the impact of pslG upon divided cells. Both the c-di-GMP reporter and cell-chain data point to a role of PslG in disposition of progeny. These results included in Figures 4, 6, and 7 are fascinating but require further probing to actually explain the role of PslG.

We thank the Reviewer for the comment and suggestion. To further understand the role of PslG, we have performed more experiments on the PslG with mutations at the glycoside hydrolytic activity sites. The results indicate that the glycoside hydrolytic activity of PslG is critical for all the phenotypes we have shown in this manuscript, suggesting an important role of the hydrolytic activity of PslG in determining the localization of PSL and its signaling function, which affects the intracellular c-di-GMP level. In the revised manuscript, corresponding results have been added in Figure 1, Figures 3-7 and associated supplement figures. We have also added several sentences in the discussion at L338 –L348.

A deeper interrogation of cell division may uncover a direct connection between PslG and the cell cycle. Several other types of experiments may be useful to discern this connection.

We thank the Reviewer for the comment. Our results show that the long bacterial chains can be disconnected within a few minutes by addition of purified PslG in media (Movie S2 and S3), suggesting that bacteria cells in the chains are actually completely divided daughter cells which are connected by PSL on bacterial periphery. In addition, Measurements of growth curves as well as the measured total biomass of matured biofilms are all similar between WT and *pslG* deletion mutants. Taken together these results, it seems that the cell division in long chains is very likely not directly connected with PslG. PslG may rather indirectly affect the intracellular c-di-GMP level and the cell cycle through changing the PSL signaling and localization.

In the revised manuscript, we have modified text accordingly (L260-268 and L339-346).

The manuscript misses the opportunity to reference and build upon recent work by the Wozniak and Parsek labs on Pel and Psl (from just the last 6 years).

Following the reviewer’s suggestion, in the revised manuscript, we have added text on PEL and alginate in the introduction part (See L61-L66) and cited four more references, which are listed below:

Jennings LK, Dreifus JE, Reichhardt C, Storek KM, Secor PR, et al. 2021. *Pseudomonas aeruginosa* aggregates in cystic fibrosis sputum produce exopolysaccharides that likely impede current therapies. *Cell Rep* 34: 108782.

Jennings LK, Storek KM, Ledvina HE, Coulon C, Marmont LS, et al. 2015. Pel is a cationic exopolysaccharide that cross-links extracellular DNA in the *Pseudomonas aeruginosa* biofilm matrix. *Proc Natl Acad Sci U S A* 112: 11353-58

Baker P, Hill PJ, Snarr BD, Alnabelseya N, Pestrak MJ, Lee MJ, Jennings LK, Tam J, Melnyk RA, Parsek MR, Sheppard DC, Wozniak DJ, Howell PL. 2016. Exopolysaccharide biosynthetic glycoside hydrolases can be utilized to disrupt and prevent *Pseudomonas aeruginosa* biofilms. Science Advances, 2:e1501632. DOI: 10.1126/sciadv.1501632, PMID: 27386527

Ma L, Wang J, Wang S, Anderson EM, Lam JS, Parsek MR, Wozniak DJ. 2012. Synthesis of multiple *Pseudomonas aeruginosa* biofilm matrix exopolysaccharides is post-transcriptionally regulated. Environmental Microbiology 14:1995-2005. DOI: 10.1111/j.1462-2920.2012.02753.x, PMID: 22513190

Line 116. From all of the descriptions here, the methods, and the figure 1 legend, it is not clear what has been quantified in Figure 1A as the measure of Psl.

We apologize for not expressing the legend clearly in our previous manuscript. What shown in Figure 1A are PSL production measured by dot blotting and the results of initial attachment assay in microtiter dish well of tested strains.

In the revised manuscript, we have modified both the figure legend of Figure 1 and description text (L118-L134) in the revised manuscript.

Line 131. Even though the WFPA801ΔpslG strain shows a "statistically significant" decrease in swim zone diameter in comparison to WT or un-induced conditions, the difference is minor. Certainly, this strain exhibits flagellar motility. Given the small difference in flagellar motility, which is used here as an indirect assessment of flagellar function that may apply to surface attachment, it would be more prudent to conclude "…the defect of ΔpslG on initial attachment MAY BE the result of multiple contributING FACTORS."

We thank the reviewer for the comment. We agree with what the reviewer suggested. In the revised manuscript, the corresponding text has been modified to be: “These results showed that pslG deletion did not affect the function of flagella and T4P directly, yet increasing PSL production in PBAD-pslΔpslG attenuated the swimming ability, which might impact its attachment phenotype.” (L45-L148).

Line 173. "…by changing the surface exploration at early stages of biofilm formation…" I disagree this is what the data show here and suggest this contradicts the data discussed lines 157-164 in the same paragraph where surface coverage differences are not apparent between pslG and PAO1 at N<10,000. Indeed, it would be the impact of pslG upon the 10,000-100,000 range of visits that begs for further discernment. Or, what is the working definition of "early stages" here?

We apologize for not expressing our points clearly. Here, “early stages” is not referred to the period of N<10,000, but referred to the time period from the beginning of bacterial tracking (just after the inoculation of bacterial cells) to the time point when first microcolonies in the field of view appear, which is the same as that defined in the reference (Zhao et al., 2013). Following the five stages of biofilm development defined in an earlier work (D. Monroe, PLoS Biol. 2007), which includes (I) initial attachment, (II) irreversible attachment, (III) maturation I, (IV) maturation II and (V) dispersion, the time period defined by “early stages” here covers stages (I), (II) and (III).For the data shown in Figure 3A,B,C,in terms of number of bacterial visits, “early stages” cover the whole range of N ≤ ∼100,000.

In the revised manuscript, in order to make our point more clearly and to avoid possible misunderstandings as the Reviewer pointed out, we have modified the text "…by changing the surface exploration at early stages of biofilm formation…" to "…by changing the surface exploration during microcolony formation" (L204-L205).

Line 228. The Raman spectra of the three strains clearly show many differences in presence and intensity of numerous peaks between 800-1800 cm-1. In the absence of more specific and supportive references and other controls, it is not clear that the 865 cm-1 peak is the definitive feature between these three spectra nor is it clear this peak allows for the specific discernment of a 1,4-glycosidic linkage polysaccharide. Lastly, it is not clear how the authors can even discern Psl from Pel polysaccharide by Raman given the data presented here. There may be no "structural change in Psl" observed here but rather a shift from Psl to Pel.

We agree with what the reviewer pointed out. To make a definitive conclusion from Raman data, we do need more controls and more specific and supportive references, which are beyond the scope of this study. Given such considerations as well as to be more focused on the role of PslG, in the revised manuscript, we have removed the Raman spectra data to leave them for a future work and added more data on the effect of PslG on PSL signaling (Figure 7 and corresponding description at L310-332).

Line 238. These results are difficult to interpret. If the ΔpslG strain makes less Psl, then how is the chain-forming phenotype dependent upon increased Psl production in the WFPA801ΔpslG strain?

The chain-forming phenotype of Δ*pslG* requires PSL production but does not need increased PSL production. Therefore, Δ*pslG* can form long chains even if it produces less PSL than PAO1. In addition, PSL produced by WFPA801 Δ*pslG* (named as PBAD-psl Δ*pslG* in the revised manuscript) under 0.5% arabinose is similar to the level of PAO1 as shown in Figure 1B. But under a condition of 0.1% arabinose where PSL production of PBAD-psl Δ*pslG* strain is presumably less than that under 0.5% arabinose, long chains were also observed (See Figure 5—figure supplement 2). However, under a condition of 0% arabinose where there is no PSL production, we didn’t observe long cell-chains (see Figure 4—figure supplement 1).

Reviewer #2:*Pseudomonas aeruginosa* is an opportunistic pathogen that is one of the leading causes of bacterial infection worldwide. The ability of this bacterium to form biofilms, which are aggregates of these microorganisms encased in a protective layer of polymers, is key for its ability to survive in different environments and cause disease. In this report, the role of a glycoside hydrolase enzyme, PslG, which is co-produced with more than a dozen other enzymes used by the bacteria to build their protective biofilm polymeric matrix, is investigated. The authors provide evidence that this enzyme determines cell fate in the early stages of *P. aeruginosa* biofilm formation by enabling cell detachment post-division. Understanding the natural functions of these enzymes is important because they comprise an emerging class of experimental therapeutics that target biofilms.The data presented by the authors in the manuscript are intriguing. However, the paper needs to be more carefully written because the impact will be tempered by the presentation. Understanding the natural role of glycoside hydrolases in biofilm development is a fundamental problem for the field that microbiologists have been pondering for more than a decade. Most of the conclusions are supported by data, but findings are not communicated as effectively as they could be, which detracts from what otherwise is a very nice dataset. There are some controls that are missing as well.

We thank Reviewer #2 for constructive comments. We have modified both texts and figures to improve the manuscript. More experimental data sets including controls are also added.

Key constructive criticismsThe authors should complete a complementation analysis for pslG in all their assays. As the authors note, the pslG gene is in an operon and so polar effects are a concern, and because pslG mutations appear pleiotropic, ensuring that phenotypes do not result from second site mutations elsewhere in the chromosome of engineered strains seems especially important.

We thank the Reviewer for suggestions. We have added several datsets of complementation analysis for the *pslG* deletion mutants, which include PslG expressed from plasmid under P_BAD_ promoter, P_BAD_-*pslG* inserted in chromosome attB site (Δ*pslG* attB::P_BAD_-*pslG*) and *pslG* knocked into the Δ*pslG* mutants at the original location of *pslG* (Δ*pslG*:: *pslG*). The results show that all phenotypes of Δ*pslG* can be complemented. These results have been added in Figures 1-6 and corresponding supplement figures in the revised manuscript.

To ensure that the hydrolase function of PslG is responsible for the observed phenotypes, it would be prudent to repeat some key assays in which the active site residues of PslG have been disrupted by site-directed mutagenesis.

Following reviewers’ suggestion, using strains in which the active sites of PslG are mutated (E65Q, E276Q), we have performed more tests on assays of PSL production, attachment, swimming motility, the formation of microcolony and long bacterial chains. The results indicate that the hydrolytic activity of PslG is critical for all these phenotypes (see Figures 1-7 and corresponding supplement figures).

Line 110-121. It is not clear what the authors are measuring and reporting in Figure 1A. In the text, the authors make statements about the quantities of ePSL produced by the different strains. However, the data in Figure 1 has no units and it is not clear whether the authors are measuring ePSL directly, or if the data represents something else. Could the authors quantify biofilm formation shown in the pictures below Figure 1A? Also, it would be highly beneficial if a semi-quantitation of the dot blots shown was provided too. Corresponding text should be added to the figure legend to succinctly explain the data presented in the figure.

Thanks for the suggestions. We apologize for confusing. What shown in Figure 1B are PSL production and attachment of tested strains. Results of PSL production shown in Figure 1B are semi-quantitation of the dot blots that are normalized to the level of PAO1. In the revised manuscript, we have modified the legend of Figure 1B (L774-786). We have also added the measurement results of corresponding biofilm biomass (OD560) in the revised Figure 1B.

Line 134-135. Could the authors better explain how the data led to their conclusion that PslG modulates surface attachment through multiple mechanisms? It is not clear from the text why the authors conclude that changes in motility are causally linked to defects in bacterial surface attachment, because the flagellum can be dispensable for biofilm formation depending on the assay. Also, the evidence here indicates that pslG mutations are pleiotropic.

The reviewer is right that *the flagellum can be dispensable for biofilm formation depending on the assay*. But based on our results and literature results, flagellum can also help the surface attachment of cells. At attachment stage, loss of flagella (such as ΔfliC mutant) can have a biofilm phenotype as that of a PSL-negative strain (Ma et al. 2006 J of Bacteriol). Given such considerations, since *pslG* deletion mutants show defected attachment, we think it is reasonably to check whether the swimming motility has been affected or not in these strains.

To make our point clearer, in the revised manuscript, we have modified the sentence to be: “These results show that *pslG* deletion does not affect the function of flagella and T4P directly, yet increasing PSL production in PBAD-pslΔpslG attenuated the swimming ability, which might impact its attachment phenotype. ”

Lines 228-234 and Figure 5 C. Caution needs to be applied using Raman spectroscopy on whole biofilms because the composition of biofilms is complex. How can the authors be certain that changes at 865 cm-1 correlate specifically with a glycosidic link in ePSL? There has been a demonstration that deletion of pslG affects intracellular c-di-GMP which influences many things in physiology including multiple components of the *P. aeruginosa* biofilm matrix. The authors would need to purify PSL from the WFPA801 and WFPA801 pslG strains and analyze that material to make these conclusions with confidence.

We thank Reviewer #2 for the comment. As in our reply to a similar comment of Reviewer #1, we agree with reviewers that to make a definitive conclusion from Raman data, more controls and more specific and supportive references are needed, which are beyond the scope of this study. Reviewer #2 also suggested to use purified PSL for the Raman examination. While we totally agree this is worth to try, it is not clear for us to what extent that such measurements using purified PSL could resemble those using fresh biofilms in which PSL is in its natural state, as the purification process may or may not change the properties of PSL. That is partially the reason why we examined the Raman spectra of biofilms directly in this work. In a future work, to better reveal the properties of PSL, measurements of PSL in both ways (i.e., either in biofilms or in purified state) may need to be performed.

Given such considerations as well as to be more focused on the role of PslG, in the revised manuscript, we have removed the Raman spectra data to leave them for a future work and added more data on the effect of PslG on PSL signaling (Figure 7 and corresponding description at L310-332).

Lines 212-227 and Figure 5A and B. Why isn't data for the pslG mutant also included in Figure 5A?

Thanks for the suggestion. In the revised manuscript, we have included the data of *pslG* deletion mutant and *pslG* with hydrolytic active site mutation (see Figure 6 in revision, referred to figure 5 in the previous version).

Lines 276-277. In this case, "data not shown" is not acceptable because the WFPA801 and WFPA801 pslG are engineered to have artificially high levels of PSL production. The data that are not shown correspond to key information corroborating physiological relevance to the wild type background. Please add these data to the manuscript.

Following the reviewer’s suggestion, we have added the data in the revised manuscript as Video 3.

Reviewer #3:In this manuscript, the authors investigated the role of PslG, a *Pseudomonas aeruginosa* glycosyl hydrolase, in biofilm formation and cell physiology of a *P. aeruginosa*. PslG is encoded on the 15 gene psl operon, and a previous study indicated that deletion of pslG caused reduced ePSL (extracellular PSL polysaccharide) production. However, in the previous study, the pslG deletion may have been polar on other downstream psl genes and may have affected ePSL production. Therefore, in this study the investigators constructed a pslG deletion in a strain where the entire psl operon is controlled by the arabinose-inducible promoter. In this way, ePSL is produced but the operon lacks pslG. The investigators showed that in this strain, ePSL is produced at levels similar to the wild-type strain. However, the pslG deletion strain had several phenotypic defects that suggested a functional role of the glycosyl hydrolase. Among the defects was a reduction in biofilm formation, even when ePSL was produced, and a reduction in swimming motility. The investigators also used microscopic tracking of the cells, and noticed a reduction in surface area covered, likely due to clumping of the cells. The clumping of the cells was due to their formation of cell chains, which were not observed in the wild-type strain or in the new strain, when ePSL production was not induced with arabinose. Interestingly, the investigators noted that in the cells that were unable to separate following division (cell chains) had ePSL accumulated at the boarders between the cells. The chains could be released by adding exogenous PslG enzyme to the cultures. The investigators suggested that in the absence of PslG, the ePSL may accumulate at the cell boarders, holding the daughter cells together. In addition to these phenotypes, the investigators also noticed a difference in the production of the cyclic-di-GMP signaling molecule in the pslG mutant strain. Since c-di-GMP is an important signaling molecule for biofilm production, the investigators put forth a model for the role of PslG in c-di_GMP signaling in the development of *P. aeruginosa* biofilms.Overall, I think this is a thorough study and the results are well presented.

We thank Reviewer #3 for his/her positive comments.

A weakness of this study (in my opinion) is that while adequately exploring the phenotypes of the PslG mutant in an ePSL producing strain, a functional role for PslG in polysaccharide degradation during ePSL biosynthesis is not adequately developed. The model in Figure 8 (and the abstract) implies that the role of PslG is through c-di-GMP signaling molecule. However, the data in Figure 4B and Figure 5B show only a minor effect of the pslG deletion on c-di-GMP signaling. In Figure 4B, most cells are classified as "none-bright" in both the experimental and control strain. Only a minor percentage of cells are classified as "two-bright". The difference in "two-bright" between the control and experimental in Figure 4B, although statistically different, is not a high percentage of the total cells. Figure 5A shows a moderate difference in c-di-GMP production as indicated by a reporter gene. However, Figure 5A does not include a growth curve, so the difference seen in fluorescence could be due to a difference in cell numbers/growth. This would not be surprising, given the many physiological defects of the pslG mutant. (a growth curve needs to be shown to determine if the fluorescence difference is due to cell growth differences). The model in Figure 8 implies that c-di-GMP signaling (and the "two-bright" daughter cells) is the primary signaling event for biofilm formation in the pslG mutant. In my opinion, this is overstated, given the minor effect of the pslG mutation on c-di-GMP production. As far as I know, PslG does not have a c-di-GMP binding domain, so it is difficult to draw a molecular link between the pslG mutant and c-di-GFP. The slight increase in c-di-GMP in the mutant cells may be due to an indirect effect of an overall stress response in the mutant cell, and not a direct signaling mechanism.

We thank the reviewer for his/her comments.

– We have measured the growth of strains, and the results show a similar growth of WT, *∆pslG*, and ΔP_psl_. This is also consistent with the result that *∆pslG* has similar biomass as that of WT strain shown in Figure 2. The results of growth curves have been added in Figure 7A and C in the revised manuscript (Figure 5 in the original version has been modified and renumbered to be Figure 7 in the revised manuscript). Based on the growth measurement, we can conclude that the observed differences in c-di-GMP in Figure 7 is not due to the growth difference of strains. In addition, we have also examined the total protein of stains in WFPA801 (which has renamed as P_BAD_-*psl* in this revision) background, the results indicated that P_BAD_-*psl*, P_BAD_-*psl*Δ*pslG*, and P_BAD_-*psl*Δ*pslG*::*pslG*_E165Q, E276Q_ have similar growth rate at the condition with or without arabinose.

– We agree with the reviewer on that “*the increase in c-di-GMP in the mutant cells may be due to an indirect effect … and not a direct signaling mechanism*.”, as the reviewer pointed out that “*PslG does not have a c-di-GMP binding domain”*. In fact, in Figure 8, we didn’t mean what the reviewer commented that “*the role of PslG is through c-di-GMP signaling molecule.”* nor mean that “*c-di-GMP signaling (and the "two-bright" daughter cells) is the primary signaling event for biofilm formation in the pslG mutant.*” What we would like to convey in Figure 8 is that PslG can affect the phenotype and c-di-GMP levels through PSL-mediated pathways, which then affect biofilm development.

– Psl has been shown to be able to act as a signaling substance to stimulate biofilm formation through affecting c-di-GMP (Irie et al., 2012). So by modifying Psl, PslG can indirectly affect c-di-GMP levels, which we believe is what has been happening in this study. There are multiple lines of evidence to support this. First, PslG can degrade PSL, which has been shown in literature. Second, in this study, the results of *∆pslG* and *pslG* with hydrolytic active site mutation indicate that PSL without the cutting of PslG shows a stronger signaling property (see Figure 6 and Figure 7A and B). In addition, we also examined the signaling property of PSL with or without PslG treatment. As shown in Figure 7C and D, PslG treatment significantly reduces the signaling property of PSL.

– In the revised manuscript, to better convey our point, we have modified Figure 8 to be clearer on the role of PslG by modifying the text to be “shapes PSL localization” and “enhances PSL signaling function” in the figure.

– In this study, after we checked the c-di-GMP levels of two daughter cells after cell division, the results show that there is an increase in the proportion of the case of “two-bright”, although the value of two-bright case is still minor compared to that of none-bright case. But even the number of two-bright case is small, they can play an important role in the microcolony formation. Earlier work has shown that cells form a microcolony through a PSL-based rich-get-richer mechanism (Zhao et al., 2013), during which founder cells can be very important. The cells of two-bright case have high c-di-GMP levels and thus can act as founder cells to promote microcolony formation. Therefore, the slight change in c-di-GMP levels of daughter cells in *∆pslG* can affect the microcolony formation.

I recommend that the authors read and cite the publication, "Role of an alginate lyase for alginate transport in mucoid *Pseudomonas aeruginosa*" Infect Immun. 2005 Oct;73(10):6429-36.doi: 10.1128/IAI.73.10.6429-6436.2005.In the discussion, the authors state, "To the best of our knowledge, the ability of aglycosyl hydrolase to affect bacterial phenotype and c-di-GMP levels has not beenreported until this work. It would be also very interesting to see in the future whetherother glycosyl hydrolases in different bacterial species can have similar functions."

We thank the Reviewer for providing the reference. In the revised manuscript, we have added this reference. We have modified the sentence as “To the best of our knowledge, the ability of a glycosyl hydrolase to affect c-di-GMP levels has not been reported until this work”*.*

In the paper above, the authors characterized another glycosyl hydrolase from *Pseudomonas aeruginosa*, that degrades alginate, and is encoded on the alginate biosynthetic operon (so, a very similar model system). In that study, the authors found that deletion of algL in an alginate producing strain resulted in cell death, due to accumulation of alginate in the periplasm. There is a precedence for these types of studies. The mechanism for PslG may be different from AlgL (and the PslG deletion obviously doesn't kill the bacteria), but the AlgL study could provide a hypothesis on the molecular role of PslG. In addition, both AlgL and PslG reside in the periplasm (assuming the model in reference: Frontiers in Microbiology 2:167 is correct). The authors of the current study only mention the periplasmic location for PslG in the Discussion section. The implication in the present study is that PslG is secreted. If PslG is secreted, there should be evidence for that (or a citation that shows that it is secreted). The localization of this enzyme could be crucial in understanding its function.

We thank the reviewer for his/her thoughtful comments. In the revised manuscript, we have added more discussions on *algL*. Please find at L405 – L414: “*algL* is a gene within the alginate synthesis operon of *P. aeruginosa* (Franklin et al., 2011). Interestingly, *algL* also encode a lyase to degrade alginate. An early work showed that Δ*algL* in a CF isolate mucoid strain FRD1 resulted in cell death, due to accumulation of alginate in the periplasm (Jain et al. 2005). However, a later work showed that Δ*algL* in a PAO1-derived mucoid strain PDO300 did not cause cell death, yet increase alginate production (Wang et al. 2016). *pelA* is the first gene in PEL synthesis operon, which also encodes a hydrolase (Franklin et al., 2011) that has been shown to inhibit biofilm formation as that of PslG (Baker et al., 2016). It would be interesting to investigate whether *algL* or *pelA* affects bacterial intracellular c-di-GMP levels”.

Regarding on the localization of PslG, in an earlier work of Wu et al. 2019 published in *MicrobiologyOpen*, results have shown that PslG is localized mainly in the inner membrane and some in the periplasm. PslG is only released into extracellular when bacteria are dead. Such information has been added in the introduction part in the revised manuscript (L78-80 and L88-93).

[Editors’ note: what follows is the authors’ response to the second round of review.]

The manuscript has been improved but there are some remaining issues that need to be addressed, as outlined below:Reviewer #1:The manuscript seeks to delineate the role of pslG, a glycosyl hydrolase encoded within an operon that enables the production of the Psl polysaccharide of the bacterium *Pseudomonas aeruginosa*. The results in the manuscript show numerous effects using a combination of pslG mutants and over-expression strains upon *P. aeruginosa* behavior important to colonization and biofilm development. This adds to our understanding of the multi-functional enzymes involved in regulating polysaccharide production and the overall behavior of *P. aeruginosa*.The manuscript primarily presents phenotypic results to shape their story and conclusions. Given the importance of both Psl and Pel to exopolysaccharide-dependent behaviors in *P. aeruginosa* PAO1 (and many other strains), the potential importance and contribution of Pel remains essentially ignored in this manuscript. Because both Pel and Psl can generally influence several of the phenotypes explored in this manuscript, the authors' desire to draw specific conclusions from the effect of PslG cannot be separated from potential contributions of Pel polysaccharide in these experiments.

This comment is the same as the one in the previous round. As we stated in our replies in the previous round “Based on literature results (for example, Colvin et al, 2012, Environmental Microbiology), PAO1 is a PSL-dominant strain, whose biofilm formation is mainly dependent on PSL. Thus, we speculate that PEL may have little contribution to the phenotypes of pslG mutants observed in this work.” More importantly, we constructed and tested PEL negative strains in either PAO1 or PslG mutant background, and found that all phenotypes of pslG mutants are still observed in PEL-negative strains. These results have already been shown in Figure 1-Figure supplement 1C, Figure 3-Figure supplement 3, and Figure 5-Figure supplement 3 in the previous version of manuscript. Taken together, all these results suggest that Pel polysaccharide has little contribution on the phenotypes observed in this study (the corresponding text can be found at L277-282 in the previous revision, L280-288 in this revision). Therefore, we could not agree with the comment about “the potential importance and contribution of Pel remains essentially ignored in this manuscript.”.

The story is certainly complicated as study of pslG mutants does affect levels of Psl polysaccharide.

Our very first experiment shown in Figure 1A is to tell that the level of PSL polysaccharide is not likely the reason to cause those phenotypes of *pslG* mutants. In the revised manuscript, to make this point clearer, we have revised the subtitle as “Δ*pslG* strains cannot form rings on microtiter dish wells even when PSL production is induced to the wild type level.” (L116-117).

The most compelling data in the manuscript involves the impact of pslG upon divided cells. Truly fascinating.

We thank the reviewer for the positive comment!

However, it is not clear that the authors demonstrate the importance of PslG to "normal" cell division and/or how Psl polysaccharide might contribute to cell division in the wild-type. There is no basis in the literature to suggest that pslG mutations would be common in select environments.

We think PSL polysaccharide might not interfere with “normal” cell division in wild type. It may affect cell division only in some circumstances where PSL could not be hydrolyzed by PslG and hence cannot be released properly, such as in pslG deletion background.

The reviewer is right that there has been no literature so far (to the best of our knowledge) to suggest that pslG mutations would be common in select environments. The E165Q, E276Q mutation of PslG is a mutation that causes PslG to loss its hydrolase activity. We used this mutant to further confirm the importance of the cutting of PslG during PSL polysaccharide synthesis.

Several new lines of experimental evidence have been added to the manuscript in this revised form and these changes are a great collective improvement-but the true role(s) and effect of PslG are still not clear from the evidence presented.

We thank the reviewer for the comment. Here it is not clear for us what the reviewer means by saying “true role(s)”.

In this study, we showed that lacking of PslG or its hydrolytic activity in PAO1 enhances the signaling function of PSL, changes the relative level of cyclic-di-GMP within daughter cells during cell division and shapes the localization of PSL on bacterial periphery, thus results in long chains of bacterial cells, fast-forming biofilm microcolonies. We believe these results have revealed the important roles and effects of PslG on the cell morphology and cell activities such as biofilm development. However, to reveal the exact mechanisms at the molecular level, more studies are needed, and we hope our work can inspire more future studies in this direction.

There are several sentences where the English syntax and grammar require editing.

We thank the reviewer for the comment. We have checked through the manuscript and corrected all grammatical errors that we found.

Line 47: add “an”;

Line 47: “protect” change to “protects”;

Line 147: “increasing” change to “inducing”;

Line 180: “hierarchical” change to “non-uniform”;

Line 188: “more microcolonies of Δ*pslG* are formed in the field of view” change to “Δ*pslG* formed more microcolonies in the field of view”;

Line 192: add “namely”;

Line 194: delete “more”;

Line 219: “affects” change to “affect”;

Line 233-Line 234: “with arabinose induction” change to “when PSL production was induced with arabinose”; Line 295: delete “a”;

Line 296: delete “a”;

Line 295: “level” change to “levels”;

Line 521: add “and”;

Line 571: “abacterial” change to “a bacterial”;

Line 572: “. Otherwise not” change to “, otherwise not”;

Line 159. There does not appear to be fewer bacterial cells in the pslG biofilms-and no quantification is given to support this statement.

From Figure 2D, it can clearly be seen that there are less green fluorescence signals in pslG mutant biofilms than in PAO1. As the fluorescence signal is from each bacterial cell, less fluorescence signal in pslG mutant biofilms corresponds to fewer bacterial cells in it. As this is qualitatively clear, we don’t think it is necessary to do the quantitative measurement for this particular statement. In addition, for quantitative measurements, we have shown the total biomass and thickness in Figure 2B. The results show that pslG mutant has a higher maximum thickness, yet a similar total biomass compared with PAO1. Thus, we can deduce that on average pslG mutant would have less number of bacterial cells than PAO1 for each section of biofilms (such as shown in Figure 2D), which is consistent with the qualitative observation of Figure 2D.

Line 178. "…a more hierarchical bacterial visit distribution for pslG…" It is not clear how this shows hierarchical visitors or a broad range of visit frequencies. How is this description evident based upon what is shown in Figure 3B?

We thank the reviewer for the comment. In Figure 3B, the number of bacterial visits is color-coded, with cyan color standing for 1000 bacterial visits and black color standing for 0 bacterial visit. Comparing the bacterial visit distribution between PAO1 and D*pslG* mutant, we can see that the bacterial visit distribution is more non-uniform in D*pslG* than in PAO1; Under the same total bacterial visits (*N*~ 100000), D*pslG* displays several concentrated patches with green-yellowish to cyan-like which stand for about 200~1000 bacterial visits and large blackish regions which have near zero bacterial visits, while PAO1 displays a large portion of surface with red-like colors which stand for ~ 100 bacterial visits. Quantitatively, the maximum of bacterial visit number for D*pslG* is higher than that in PAO1, and the power law exponent of bacterial visit distribution is also less negative for D*pslG* than for PAO1, so D*pslG* has a more hierarchical bacterial visit distribution. The same method has been used in earlier work (Kun, Zhao, Boo, Shan, Tseng, & Bernard, et al. (2013). Psl trails guide exploration and microcolony formation in *Pseudomonas aeruginosa* biofilms. Nature, 497(7449), 388-391).

In the revised manuscript, we have modified the text to be: “…a more non-uniform bacterial visit distribution for Δ*pslG* (L180)”. In the legend of Figure 3B, we also added the maximum bacterial visit number for each bacterial visit distribution map.

Line 184. There appear to be 27-28 cells in the PAO1 cluster detailed in Figure 3D.

We thank the reviewer for the comment. In this study, microcolonies are defined as clusters of more than 30 cells. For clusters, we used a minimum distance criterion to judge whether a cell belonged to a cluster or not. If the minimum distance between any point of the scrutinized cell body and any point of any cell body of the cluster, is smaller than 0.5µm (i.e., about one width of a bacterial cell), then the scrutinized cell is considered to belong to the cluster, otherwise not. Based on these definitions, the PAO1 cluster shown in Figure 3D has 30 cells and thus is a microcolony.

To illustrate these cells clearly, in Author response image 1, the cells are colored manually in a way that neighboring cells don’t have the same color so that we can easily count how many cells that have been involved in this cluster.

**Author response image 1. sa2fig1:** 

Line 188. "…microcolonies of ΔpslG are more compact". This representation of the data in Figure 3F does not necessarily support this statement. Is this a projection view showing a 2D image of the thicker 3D biofilms as described in Figure 2? How was this quantified?

We thank the reviewer for the comment. The images shown in Figure 3F were taken after 10 hours’ growth in a flow cell, while those shown in Figure 2 are confocal laser scanning microscope images of pellicles grown on air-liquid interface. So they are different. Regarding on the images shown in Figure 3F, they are representative images taken by a regular fluorescence microscope to give a general view of microcolonies formed by PAO1 and Δ*pslG*. They can be considered as projection views of a slab sample (the thickness of the slab is closely related to the depth of field of the equipment, which is about 0.54 micron under our set-up). We do agree with the reviewer that this statement may not appropriate. In the revised manuscript, we have modified the sentence to be “Tracking GFP-tagged bacteria in a flow cell also showed that the PAO1 biofilms covered more surface (Video 1 and Figure 3F) and ΔpslG cells tended to form microcolonies with strong fluorescence intensity (Video 2 and Figure 3F), suggesting that bacteria were packed within microcolonies. This is consistent with the results visit distribution map shown in Figure 3B.” (L208-L212). We also included representative videos (Video 1 and 2) tracking the microcolony formation of PAO1 and Δ*pslG* in this revision.

Line 221. How were these twitching speed experiments performed? These are not described in the supplemental Figure legend or the methods.

We thank the reviewer for the comment. The twitching speed of each tracked cell at frame n was calculated by the displacement of the cell between n^th^ and (n+1) ^th^ frames divided by the corresponding time interval. In the methods, we described how the experimental data were collected (lines 499-508) and briefly how the single cell tracking image analysis was performed (lines 544-549). As such imaging analysis methods have been described in detail in earlier studies (Zhang et al., 2018, and associated references therein), we did not describe it in details in the original version of the manuscript.

In the revised manuscript, we have added the above description on the measurement of twitching speed: “Specifically, the twitching speed of each tracked cell at frame n was calculated by the displacement of the cell between n^th^ and (n+1)^th^ frames divided by the corresponding time interval” (Lines 550-552).

Line 240. "…were observed most frequently in either ΔpslG or PBAD-pslΔpslG strain…" The purpose of this statement is not clear. These are the only strains shown in Figure 4. Is this a comparison?

We apologize for not expressing our point clearly. Here the purpose of this statement is to characterize the number of cells in a chain (i.e. the chain length in terms of the number of cells). Since WT cells don’t show long chains, we just show the distribution of cell chains in two pslG mutants. But we didn’t mean to compare between different strains. From the distribution of cell chains, we can see that the cell chains consisting of four cells are observed most frequently among all observed chains. This is true in both Δ*pslG* and P_BAD_-*psl*Δ*pslG* strains.

In the revised manuscript, to make our point clearer, we have modified the text to be “The length of chains varied. Among all the observed chains, the chains consisting of 4 cells were observed most frequently, which is true both in ΔpslG and in PBAD-pslΔpslG strains (Figure 4C) (L242-244).”

Line 241. What is the support for the statement "…yet some can also be broken by bending of chains"? This is from separate experimental evidence? Or just an observation?

This can be seen clearly from the Video 3. In the revised manuscript, we have cited the corresponding video in the text (L246).

Line 246. This Figure 5 data is the most intriguing of the manuscript.

We thank the Reviewer for the positive comment!

Line 309. "…which might enhance their stay on the surface, reduce their surface motility, and promote the microcolony formation". Given the small variation in the data shown in Figure 6B, support for this statement really requires specific testing. Further, not all previous data in the manuscript support it. The text in an earlier section (starting line 213) effectively concludes that there is no quantifiable motility difference between these strains.

We thank the reviewer for the insightful comment. Following that, we have modified the text to make it be more appropriate given the results that we obtained in this study. In the revised manuscript, the sentence is now to be: “During asymmetric divisions, daughter cells with high c-di-GMP levels keep staying on the surface and daughter cells with low c-di-GMP levels tend to move away (Christen et al., 2010; Laventie et al., 2019). Therefore, both daughter cells with high c-di-GMP levels might enhance bacterial stay on the surface and alter their movement pattern. In addition, a high level of c-di-GMP enhances PSL production. An earlier work has shown that cells form a microcolony through a PSL-based rich-get-richer mechanism (Zhao et al., 2013), during which founder cells can be very important. The cells of two-bright cases have high c-di-GMP levels and thus can act as founder cells to promote microcolony formation. Altogether, the slight change in c-di-GMP levels of daughter cells in pslG mutants can be likely one of reasons to promote the formation of microcolony and long bacterial chains.” (L312-323).

Line 497. The Psl staining with TRITC-HHA is specific to Psl? Is there binding to Pel or other polysaccharides?

Yes, the Psl staining with TRITC-HHA is specific to Psl. No binding to Pel or alginate. This can be found in the papers of Ma et al. PLoS Pathogens 2009 and Ma et al. FEMS Immunolology & Medical Microbiol. 2012.

Line 770. The Figure 1 title is not the best description of what is shown here. No "increasing" levels of Psl are shown. There are two. The data show adding of Psl into a pslG background.

Thanks for the comment. We have revised the text as “Inducing PSL production in ΔpslG background cannot recover its defects on bacterial initial attachment and yet affects swimming motility.”

Line 821. It is not stated anywhere in the legend or methods from how many frames the data analyzed for Figure 3E were obtained. There are error bars, but it is not clear what these represent.

We thank the reviewer for pointing this out. In the revised manuscript, we have added the corresponding information in the legend of Figure 3E: “The number of microcolonies in the field of view formed by PAO1, Δ*pslG*, Δ*pslG*::*pslG* and Δ*pslG*::*pslG*_E165Q,E276Q_ at 10 hrs after inoculation in a flow cell. The number (*N*) of frames analyzed are 14, 43, 52, 51 for PAO1, Δ*pslG*, Δ*pslG*::*pslG* and Δ*pslG*::*pslG*_E165Q,E276Q_, respectively. Error bars represent standard deviation of the mean. Statistical significances are measured using one-way ANOVE. n.s., not significant; *p < 0.05; **p < 0.001; ***p < 0.0001” (L848-851).

Line 831. The criteria for "microcolony" is not entirely clear. Some of these cells clearly have different spacing but are outlined with dashed lines.

We thank the reviewer for the comment. The definitions of microcolony (lines 185-186) and cluster (lines 568-572) have been stated in the text. The microcolonies outlined with dashed lines in Figure 3 are all determined based on these definitions. Particularly, in this study whether or not a cell belongs to a cluster is judged by a minimum distance criterion. We note that this criterion does not mean that cells need to be most densely packed inside a cluster. For example, for the outlined microcolony of P_BAD_-*psl* in the Figure 3—figure supplement 1A, each of cells along the dashed outline has at least one neighboring cell that is within the distance of 0.5µm of the cell (i.e. meets the requirement of minimum distance criterion) and belongs to the cluster. But there is a clear empty (free of cells) region enclosed by cells inside the microcolony. How dense a microcolony can be will depend on multiple factors including cell motility, cell growth and the incubation time etc.

Reviewer #2:The manuscript by Zhang and colleagues carefully describes the phenotypes of pslG mutants of *P. aeruginosa*. PslG encodes a glyocoside hydrolase. While the biochemistry of this enzyme has been understood for many years, its physiologically relevant role in *P. aeruginosa* biofilm formation has remained ill-defined.The experiments in the manuscript have been meticulously executed. There are controls and complementation analyses that provide confidence in the results obtained. The technical proficiency with microscopy is commendable.

We thank reviewer #2 for the positive comments.

However, while the authors provide a data-rich manuscript, an understanding of the consequences of PslG expression appears lacking beyond phenotyping.

We thank reviewer #2 for the comment. Our manuscript reveals at least one mechanism beyond phenotyping, which is the hydrolysis of PslG on PSL during biosynthesis can change the signaling function of PSL and shape the localization of PSL on bacterial periphery, leading to a higher portion of bacterial cells that are not able to go asymmetric divisions, which may become the founder cells for microcolonies formation, and finally resulting in long bacterial chains as well as the change of bacterial surface explore patterns. We have added several sentences to describe this point (See L321-323). We have also revised the last paragraph of result section (L312-323).

Perhaps this criticism is most pertinent to the observed changes in the power-law distribution for bacteria during the earliest stages of biofilm formation for pslG mutants. Interpretation/experimentation is absent that connects these observations to social biology. Such connections, which are front and center in prior work published by one of the co-authors (Dr. Khun Zhao), could help to explain conservation of glycoside hydrolases among synthase-dependent exopolysaccharide secretion systems like the Psl synthase. For example, beyond the careful phenotyping presented in this paper, co-culture of Pbad-Psl with Pbad-Psl-PslG strains, or perhaps wild type and pslG strains, that have been uniquely labelled with fluorescent proteins and tracked using Dr. Kun Zhao's elegant single-cell methods could directly demonstrate fitness changes for pslG mutants in surface exploration or colonization relative to wild type. In principle, wouldn't such a fitness cost provide an explanation for PslG function that is rooted in social evolutionary theory? Perhaps there are some trade-offs that aren't yet apparent. The link to c-di-GMP signaling provides some molecular insight even the sensory perception and signal transduction pathway is not yet fully known. Such analyses could take the work assembled here to the next level with little additional experimental effort, and as such, strikes me as a missed opportunity to provide significant, additional understanding of some really nice data.

We thank Reviewer #2 for the suggestions. It is certainly a great idea, which might lead to a new story. Our manuscript is already data-rich as reviewers stated above, thus adding more data may just dilute the main information.

Following the suggestion, we have tried once the co-culture experiment of PAO1 and pslG mutant. Some preliminary results are shown in Author response image 2. PAO1 are GFP-tagged while Δ*pslG* is not, so that they can be differentiated from each other.

**Author response image 2. sa2fig2:** (**A**): The biofilm microcolonies formation by PAO1 with Δ*pslG* in flow cell. PAO1 are GFP-tagged while Δ*pslG* is not. (**B**): Surface coverage maps at a total of 10000, 50000 and 100000 bacterial visits for PAO1 with Δ*pslG*, PAO1 in co-culture and Δ*pslG* in co-culture. (**C**): The visit frequency distributions of PAO1 with ΔpslG, PAO1 in co-culture and ΔpslG in co-culture.

However, due to the phototoxicity caused by fluorescence illumination, recording of cell behavior under fluorescence illumination cannot be performed at a frame rate that is fast enough to meet the requirement of cell tracking (currently, the fluorescence recording is at a frame rate of 1 frame per 15minutes). We have also tried to record the cell behavior through both bright-field channel and fluorescence channel with a recording frame rate of 1 frame per 3 second for bright-field channel and 1 frame per 15 minutes for fluorescence channel. However, using this method, we cannot distinguish which type of cell is for those cells whose appearance and disappearance both happen during a bright-field recording period between two consecutive fluorescence recording time points. So, new methods for distinguishing two strains and/or new analysis techniques are needed, which are beyond the scope of this manuscript.

In addition, to get conclusive results of co-culture experiments, more control experiments are also needed including the single-strain experiments under the same fluorescence illumination conditions. Considering our manuscript is already data-rich as reviewers stated above, adding more data may just dilute the main point of this manuscript.

Given these considerations, we believe the study on co-culture experiments deserves a complete and separate treatment, and it is more appropriate to leave it for future work.